# Forward Operator Estimation in Generative Models with Kernel Transfer Operators

## Abstract

Generative models which use explicit density modeling (e.g., variational autoencoders, flow-based generative models) involve finding a mapping from a known distribution, e.g. Gaussian, to the unknown input distribution. This often requires searching over a class of non-linear functions (e.g., representable by a deep neural network). While effective in practice, the associated runtime/memory costs can increase rapidly, usually as a function of the performance desired in an application. We propose a much cheaper (and simpler) strategy to estimate this mapping based on adapting known results in kernel transfer operators. We show that our formulation enables highly efficient distribution approximation and sampling, and offers surprisingly good empirical performance that compares favorably with powerful baselines, but with significant runtime savings. We show that the algorithm also performs well in small sample size settings (in brain imaging).

## 1 Introduction

Generative modeling, in its unconditional form, refers to the problem of estimating the data generating distribution: given *i.i.d.* samples $\mathbf{X}$ with an unknown distribution $P_X$, a generative model seeks to find a parametric distribution that closely resembles $P_X$. In modern deep generative models, we often approach this problem via a *latent variable* – i.e., we assume that there is some variable $Z \in \mathcal{Z}$ associated with the observed data $X \in \mathcal{X}$ that follows a *known* distribution $P_Z$ (also referred to as the *prior* in generative models). Thus, we can learn a mapping $f : \mathcal{Z} \to \mathcal{X}$ such that the distribution after transformation, denoted by $P_{f(Z)}$, aligns well with the data generating distribution $P_X$. Therefore, sampling from $P_X$ becomes convenient since $P_Z$ can be efficiently sampled. Frequently, $f$ is parameterized by deep neural networks and optimized with stochastic gradient descent (SGD).

Existing generative modeling methods variously optimize the transformation $f$, most commonly modeling it as a Maximum Likelihood Estimation (MLE) or distribution matching problem. For instance, given data $\mathbf{X} = \{\mathbf{x}_1, \ldots, \mathbf{x}_n\}$, a variational autoencoder (VAE) (Kingma & Welling, 2013) first constructs $Z$ through the approximate posterior $q_{Z|X}$ and maximizes a lower bound of likelihood $p_{f(Z)}(\mathbf{X})$. Generative adversarial networks (GANs) (Goodfellow et al., 2014) relies on a simultaneously learned discriminator such that samples of $P_{f(Z)}$ are indistinguishable from $\mathbf{X}$. Results in (Arjovsky et al., 2017; Li et al., 2017) suggest that GANs minimize the distributional discrepancies between $P_{f(Z)}$ and $P_X$. Flow-based generative models optimize $p_{f(Z)}(\mathbf{X})$ explicitly through the *change of variable rule* and efficiently calculating the Jacobian determinant of the inverse mapping $f^{-1}$.

In all examples above, the architecture or objective notwithstanding, the common goal is to find a suitable function $f$ that reduces the difference between $P_{f(Z)}$ and $P_X$. Thus, a key component in many deep generative models is to learn a *forward operator* as defined below.

**Definition 1.1** (Forward operator). *A forward operator $f^\star \in \mathcal{C} : \mathcal{Z} \to \mathcal{X}$ is defined to be a mapping associated with some latent variable $Z \sim P_Z$ such that $f^\star = \arg\min_{f \in \mathcal{C}} d(P_{f(Z)}, P_X)$ for some function class $\mathcal{C}$ and a distance measure $d(\cdot, \cdot)$.*

**Motivation:** The specifics of the forward operator may differ from case to case. But its properties and how it is estimated numerically greatly influences the empirical performance

of the model. For instance, mode collapse issues in GANs are well known and solutions continue to emerge (Srivastava et al., 2017). To learn the forward operator, VAEs use an approximate posterior $q_{Z|X}$ that may sometimes fail to align with the prior (Kingma et al., 2016; Dai & Wipf, 2019). Flow-based generative models enable direct access to the posterior likelihood, yet in order to tractably evaluate the Jacobian of the transformation during training, one must either restrict the expressiveness at each layer (Dinh et al., 2017; Kingma & Dhariwal, 2018) or use more involved solutions (Chen et al., 2018). Of course, solutions to mitigate these weaknesses (Ho et al., 2019) remains an active area of research.

The starting point of our work is to evaluate the extent to which we can *radically* simplify the forward operator in deep generative models. Consider some desirable properties of a hypothetical forward operator (in Def. (1.1)): **(a)** Upon convergence, the learned operator $f^\star$ minimizes the distance between $P_X$ and $P_{f(Z)}$ over all possible operators of a certain class. **(b)** The training directly learns the mapping from the prior distribution $P_Z$, rather than a variational approximation. **(c)** The forward operator $f^\star$ can be efficiently learned and sample generation is also efficient. It would appear that these criteria violate the "no free lunch rule", and some compromise must be involved. Our goal is to investigate this trade-off: which design choices can make this approach work? Specifically, a well studied construct in dynamical systems, namely the Perron-Frobenius operator (Lemmens & Nussbaum, 2012), suggests an alternative *linear* route to model the forward operator. Here, we show that if we are willing to give up on a few features in existing models – this may be acceptable depending on the downstream use case – then, the forward operator in generative models can be efficiently approximated as the estimation of a *closed-form linear operator* in the reproducing kernel Hilbert space (RKHS). With simple adjustments of existing results, we identify a novel way to replace the expensive training for generative tasks with a simple principled kernel approach.

**Contributions.** Our results are largely based on results in kernel methods and dynamical systems, but we demonstrate their relevance in generative modeling and complement recent ideas that emphasize links between deep generative models and dynamical systems. Our contributions are **(a)** We propose a *non-parametric* method for transferring a known prior density *linearly* in RKHS to an unknown data density – equivalent to learning a nonlinear forward operator in the input space. When compared to its functionally-analogous module used in other deep generative methods, our method avoids multiple expensive training steps yielding significant efficiency gains; **(b)** We evaluate this idea in multiple scenarios and show competitive generation performance and efficiency benefits with pre-trained autoencoders on popular image datasets including MNIST, CIFAR-10, CelebA and FFHQ; **(c)** As a special use case, we demonstrate the advantages over other methods in limited data settings.

## 2 Preliminaries

We briefly introduce reproducing kernel Hilbert space (RKHS) and kernel embedding of probability distributions, concepts we will use frequently.

**Definition 2.1** (RKHS (Aronszajn, 1950)). *For a set $\mathcal{X}$, let $\mathcal{H}$ be a set of functions $g : \mathcal{X} \to \mathbf{R}$. Then, $\mathcal{H}$ is a reproducing kernel Hilbert space (RKHS) with a product $\langle \cdot, \cdot \rangle_{\mathcal{H}}$ if there exists a function $k : \mathcal{X} \times \mathcal{X} \to \mathbf{R}$ (called a reproducing kernel) such that (i) $\forall x \in \mathcal{X}, g \in \mathcal{H}, g(x) = \langle g, k(x, \cdot) \rangle_{\mathcal{H}}$; (ii) $\mathcal{H} = cl(span(\{k(x, \cdot), x \in \mathcal{X}\}))$, where $cl(\cdot)$ is the set closure.*

| Notations | | Meaning |
|---|---|---|
| $Z$ | $X$ | Random variable |
| $\mathbf{Z}$ | $\mathbf{X}$ | Data samples |
| $\mathcal{Z}$ | $\mathcal{X}$ | Domain |
| $P_Z$ | $P_X$ | Distribution |
| $p_Z$ | $p_X$ | Density function |
| $k$ | $l$ | Kernel function |
| $\mathcal{H}$ | $\mathcal{G}$ | RKHS |
| $\phi(\cdot)$ | $\psi(\cdot)$ | Feature mapping |
| $\mathcal{E}_k$ | $\mathcal{E}_l$ | Mean embedding operator |
| $\mu_Z$ | $\mu_X$ | Kernel mean embedding |

Table 1: Notations used in this paper.

The function $\phi(x) = k(x, \cdot) : \mathcal{X} \to \mathcal{H}$ is referred to as the *feature mapping* of the induced RKHS $\mathcal{H}$. A useful identity derived from feature mappings is the *kernel mean embedding*: it defines a mapping from a probablity measure in $\mathcal{X}$ to an element in the RKHS.

**Definition 2.2** (Kernel Mean Embedding (Smola et al., 2007)). *Given a probability measure $p$ on $\mathcal{X}$ with an associated RKHS $\mathcal{H}$ equipped with a reproducing kernel $k$ such that $\sup_{x \in \mathcal{X}} k(x, x) < \infty$, the kernel mean embedding of $p$ in RKHS $\mathcal{H}$, denoted by $\mu_p \in \mathcal{H}$, is*

defined as $\mu_p = E_p[\phi(x)] = \int k(x, \cdot)p(x)dx$, and the mean embedding operator $\mathcal{E} : L^1(\mathcal{X}) \to \mathcal{H}$ is defined as $\mu_p = \mathcal{E}p$.

> **Remark 1.** *For characteristic kernels, the operator $\mathcal{E}$ is injective. Thus, two distributions $(p, q)$ in $\mathcal{X}$ are identical iff $\mathcal{E}p = \mathcal{E}q$.*

This property allows using of *Maximum Mean Discrepancy (MMD)* for distribution matching (Gretton et al., 2012; Li et al., 2017) and is common, see (Muandet et al., 2017; Zhou et al., 2018). For a finite number of samples $\{\mathbf{x}_i\}_{i=1}^n$ drawn from the probability measure $p$, an unbiased empirical estimate of $\mu_{\mathcal{H}}$ is $\hat{\mu}_{\mathcal{H}} = \frac{1}{n}\sum_{i=1}^n k(\mathbf{x}_i, \cdot)$ such that $\lim_{n\to\infty} \frac{1}{n}\sum_{i=1}^n k(\mathbf{x}_i, \cdot) = \mu_{\mathcal{H}}$.

Next, we review the covariance/cross covariance operators, two widely-used identities in kernel methods (Fukumizu et al., 2013; Song et al., 2013) and building blocks of our approach.

**Definition 2.3** (Covariance/Cross-covariance Operator). *Let $X, Z$ be random variables defined on $\mathcal{X} \times \mathcal{Z}$ with joint distribution $P_{X,Z}$ and marginal distributions $P_X$, $P_Z$. Let $(l, \phi, \mathcal{H})$ and $(k, \psi, \mathcal{G})$ be two sets of (a) bounded kernel, (b) their corresponding feature map, and (c) their induced RKHS, respectively. The (uncentered) covariance operator $\mathcal{C}_{ZZ} : \mathcal{H} \to \mathcal{H}$ and cross-covariance operator $\mathcal{C}_{XZ} : \mathcal{H} \to \mathcal{G}$ are defined as*

$$\mathcal{C}_{ZZ} \triangleq \mathbb{E}_{z \sim P_Z}[\phi(z) \otimes \phi(z)] \qquad \mathcal{C}_{XZ} \triangleq \mathbb{E}_{(x,z) \sim P_{X,Z}}[\psi(x) \otimes \phi(z)] \tag{1}$$

*where $\otimes$ is the outer product operator.*

## 3 SIMPLIFYING THE ESTIMATION OF THE FORWARD OPERATOR

**Forward operator as a dynamical system:** The dynamical system view of generative models has been described by others (Chen et al., 2018; Grathwohl et al., 2019; Behrmann et al., 2019). These strategies model the evolution of latent variables in a residual neural network in terms of its dynamics over continuous or discrete time $t$, and consider the output function $f$ as the evaluation function at a predetermined boundary condition $t = t_1$. Specifically, given an input (i.e., initial condition) $z(t_0)$, $f$ is defined as

$$f(z(t_0)) = z(t_0) + \int_{t_0}^{t_1} \Delta_t(z(t))dt \tag{2}$$

where $\Delta_t$ is a time-dependent neural network function and $z(t)$ is the intermediate solution at $t$. This view of generative models is not limited to specific methods or model archetypes, but generally useful, for example, by viewing the outputs of each hidden layer as evaluations in discrete-time dynamics. After applying $f$ on a random variable $Z \in \mathcal{Z}$, the marginal density of the output over any subspace $\Lambda \subseteq \mathcal{X}$ can be expressed as

$$\int_\Lambda p_{f(Z)}(x)dx = \int_{z \in f^{-1}(\Lambda)} p_Z(z)dz \tag{3}$$

If there exists some neural network instance $\Delta_t^\star$ such that the corresponding output function $f^\star$ satisfies $P_X = P_{f^\star(Z)}$, by Def. 1.1, $f^\star$ is a forward operator. Let $\mathbf{X}$ be a set of *i.i.d.* samples drawn from $P_X$. In typical generative learning, either maximizing the likelihood $\frac{1}{|\mathbf{X}|}\sum_{x \in \mathbf{X}} p_{f(Z)}(x)$ or minimizing the distributional divergence $d(P_{f(Z)}, P_\mathbf{X})$ requires evaluating and differentiating through $f$ or $f^{-1}$ many times.

**Towards a one-step estimation of forward operator:** Since $f$ and $f^{-1}$ in (3) will be highly nonlinear in practice, evaluating and computing the gradients can be expensive. Nevertheless, the dynamical systems literature suggests a *linear* extension of $f^\star$, namely the *Perron-Frobenius* operator or transfer operator, that conveniently transfers $p_Z$ to $p_X$.

**Definition 3.1** (Perron-Frobenius operator (Mayer, 1980)). *Given a dynamical system $f : \mathcal{X} \to \mathcal{X}$, the Perron-Frobenius (PF) operator $\mathcal{P} : L^1(\mathcal{X}) \to L^1(\mathcal{X})$ is an infinite-dimensional linear operator defined as $\int_\Lambda (\mathcal{P}p_Z)(x)dx = \int_{z \in f^{-1}(\Lambda)} p_Z(z)dz$ for all $\Lambda \subseteq \mathcal{X}$.*

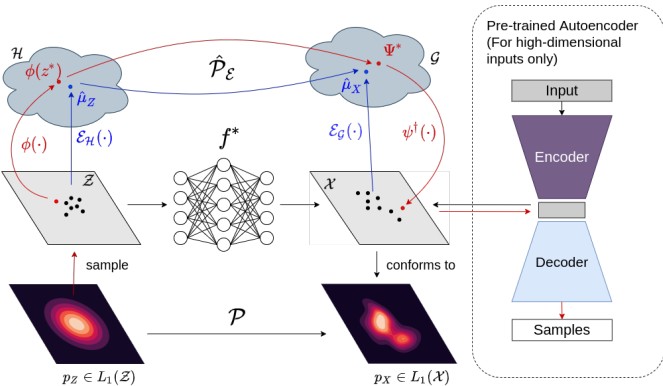

Figure 1: Summary of our framework. The proposed operator is estimated to transfer RKHS embedded densities from one to another (blue arrows). Generating new samples (red arrows) involves embedding the prior samples $z^* \sim P_Z$, applying the operator $\Psi^* = \hat{\mathcal{P}}_\mathcal{E}\phi(z^*)$, and finding preimages $\psi^\dagger(\Psi^*)$. The pre-trained autoencoder projects the data onto a *smooth* latent space and is only required when generating high-dimensional data such as images.

Although in Def. 3.1, the PF operator $\mathcal{P}$ is defined for self-maps, it is trivial to extend $\mathcal{P}$ to mappings $f : \mathcal{Z} \to \mathcal{X}$ by restricting the RHS integral $\int_{z \in f^{-1}(\Lambda)} p_Z(z)dz$ to $\mathcal{Z}$.

It can be seen that, for the forward operator $f^*$, the corresponding PF operator $\mathcal{P}$ satisfies

$$p_X = \mathcal{P}p_Z. \qquad (4)$$

If $\mathcal{P}$ can be efficiently computed, transferring the tractable density $p_Z$ to the target density $p_X$ can be accomplished simply by applying $\mathcal{P}$. However, since $\mathcal{P}$ is an infinite-dimensional operator on $L^1(\mathcal{X})$, it is impractical to instantiate it explicitly and exactly. Nonetheless, there exist several methods for estimating the Perron-Frobenius operator, including Ulam's method (Ulam, 1960) and the Extended Dynamical Mode Decomposition (EDMD) (Williams et al., 2015a). Both strategies project $\mathcal{P}$ onto a finite number of hand-crafted basis functions – this may suffice in many settings but may fall short in modeling highly complex dynamics.

**Kernel-embedded form of PF operator:**   A natural extension of PF operator is to represent $\mathcal{P}$ by an infinite set of functions (Klus et al., 2020), e.g., projecting it onto the bases of an RKHS via the *kernel trick*. There, for a characteristic kernel $l$, the *kernel mean embedding* uniquely identifies an element $\mu_X = \mathcal{E}_l p_X \in \mathcal{G}$ for any $p_X \in L^1(\mathcal{X})$. Thus, to approximate $\mathcal{P}$, we may alternatively solve for the dynamics from $p_Z$ to $p_X$ in their *embedded* form. Using Tab. 1 notations, we have the following linear operator that defines the dynamics between two embedded densities.

**Definition 3.2** (Kernel-embedded Perron-Frobenius operator (Klus et al., 2020)). *Given $p_Z \in L^1(\mathcal{X})$ and $p_X \in L^1(\mathcal{X})$. Denote $k$ as the **input kernel** and $l$ as the **output kernel**. Let $\mu_X = \mathcal{E}_l p_X$ and $\mu_Z = \mathcal{E}_k p_Z$ be their corresponding mean kernel embeddings. The kernel-embedded Perron-Frobenius (kPF) operator, denoted by $\mathcal{P}_\mathcal{E} : \mathcal{H} \to \mathcal{G}$, is defined as*

$$\mathcal{P}_\mathcal{E} = \mathcal{C}_{XZ}\mathcal{C}_{ZZ}^{-1} \qquad (5)$$

**Proposition 3.1** (Song et al. (2013)). *With the above definition, $\mathcal{P}_\mathcal{E}$ satisfies*

$$\mu_X = \mathcal{P}_\mathcal{E}\mu_Z \qquad (6)$$

*under the conditions: (i) $\mathcal{C}_{ZZ}$ is injective (ii) $\mu_t \in range(\mathcal{C}_{ZZ})$ (iii) $\mathbb{E}[g(X)|Z = \cdot] \in \mathcal{H}$ for any $g \in G$.*

The last two assumptions can sometimes be difficult to satisfy for certain RKHS (see Theorem 2 of Fukumizu et al. (2013)). In such cases, a relaxed solution can be constructed by replacing $\mathcal{C}_{ZZ}^{-1}$ by a regularized inverse $(\mathcal{C}_{ZZ} + \lambda I)^{-1}$ or a Moore-Penrose pseudoinverse $\mathcal{C}_{ZZ}^\dagger$.

The following proposition shows commutativity between the (kernel-embedded) PF operator and the mean embedding operator, showing its equivalence to $\mathcal{P}$ when $l$ is characteristic.

**Proposition 3.2** ((Klus et al., 2020)). *With the above notations, $\mathcal{E}_l \circ \mathcal{P} = \mathcal{P}_\mathcal{E} \circ \mathcal{E}_k$.*

**Transferring embedded densities with the kPF operator:** The kPF operator is a powerful tool that allows transferring embedded densities in RKHS. The main steps are:

> **(1)** Use mean embedding operator $\mathcal{E}_l$ on $p_Z$. Let us denote it by $\mu_Z$.
> **(2)** Transfer $\mu_Z$ using kPF operator $\mathcal{P}_\mathcal{E}$ to get the mean embedded $p_X$, given by $\mu_X$.

Of course, in practice with finite data, $\{\mathbf{x}_i\}_{i\in[n]} \sim P_X$ and $\{\mathbf{z}_i\}_{i\in[n]} \sim P_X$, $\mathcal{P}_\mathcal{E}$ must be estimated empirically (see Klus et al. (2020) for an error analysis).

$$\hat{\mathcal{P}}_\mathcal{E} = \hat{\mathcal{C}}_{XZ}(\hat{\mathcal{C}}_{ZZ})^{-1} \approx \Psi(\Phi^T\Phi + \lambda nI)^{-1}\Phi^T \approx \Psi(\Phi^T\Phi)^\dagger\Phi^T$$

where $\Phi = [k(\mathbf{z}_1,\cdot),\cdots,k(\mathbf{z}_n,\cdot)]$, $\Psi = [l(\mathbf{x}_1,\cdot),\cdots,l(\mathbf{x}_n,\cdot)]$ are simply the corresponding feature matrices for samples of $P_X$ and $P_Z$, and $\lambda$ is a small penalty term.

**Learning kPF for unconditional generative modeling:** Some generative modeling methods such as VAEs and flow-based formulations explicitly model the latent variable $Z$ as conditionally dependent on the data variable $X$. This allows deriving/optimizing the likelihood $p_{f(Z)}(X)$. This is desirable but may not be essential in all applications. To learn a kPF, however, $X$ and $Z$ can be independent RVs. While it may not be immediately obvious why we could assume this independence, we can observe the following property for the empirical kPF operator, assuming that the empirical covariance operator $\hat{\mathcal{C}}_{ZZ}$ is non-singular:

$$\hat{\mathcal{P}}_\mathcal{E}\hat{\mu}_Z = \hat{\mathcal{C}}_{XZ}\hat{\mathcal{C}}_{ZZ}^{-1}\hat{\mu}_Z = \underbrace{\Psi\Phi^\top}_{\hat{c}_{XZ}}(\underbrace{\Phi\Phi^\top}_{\hat{c}_{ZZ}})^{-1}\Phi 1_n = \Psi(\Phi^\top\Phi)^{-1}\Phi^\top\Phi 1_n = \Psi 1_n = \hat{\mu}_X \qquad (7)$$

Suppose that $\{\mathbf{x}_i\}_{i\in[n]}$ and $\{\mathbf{z}_j\}_{i\in[n]}$ are independently sampled from the marginals $P_X$ and $P_Z$. It is easy to verify that (7) holds for any pairing $\{(\mathbf{x}_i,\mathbf{z}_j)\}_{(i,j)\in[n]\times[n]}$. However, instantiating the RVs in this way rules out the use of kPF for certain downstream tasks such as controlled generation or mode detection, since $Z$ does not contain information regarding $X$. Nevertheless, if sampling is our only goal, then this instantiation of kPF will suffice.

**Mapping $Z$ to $\mathcal{G}$:** Now, since $\mathcal{P}_\mathcal{E}$ is a *deterministic* linear operator, we can easily set up a scheme to map samples of $Z$ to elements of $\mathcal{G}$ where the expectation of the mapped samples equals $\mu_X$

Define $\phi(z) = k(z,\cdot)$ and $\psi(x) = l(x,\cdot)$ as feature maps of kernels $k$ and $l$. We can rewrite $\mu_X$ as

$$\mu_X = \mathcal{P}_\mathcal{E}\mathcal{E}_k p_Z = \mathcal{P}_\mathcal{E}E_Z[\phi(Z)] = E_Z[\mathcal{P}_\mathcal{E}(\phi(Z))] = E_Z[\psi\left(\psi^{-1}\left(\mathcal{P}_\mathcal{E}\phi\left(Z\right)\right)\right)] \qquad (8)$$

Here $\psi^{-1}$ is the inverse or the *preimage map* of $\psi$. Such an inverse, in general, may not exist (Kwok & Tsang, 2004; Honeine & Richard, 2011). We will discuss a procedure to approximate $\psi^{-1}$ in §4.1. In what follows, we will temporarily assume that an exact preimage map exists and is tractable to compute.

Define $\Psi^* = \hat{\mathcal{P}}_\mathcal{E}\phi(Z)$ as the *transferred sample* in $\mathcal{G}$ using the empirical embedded PF operator $\hat{\mathcal{P}}_\mathcal{E}$. Then the next result shows that asymptotically the transferred samples converge in distribution to the target distribution.

**Proposition 3.3.** *As $n \to \infty$, $\psi^{-1}(\Psi^*) \xrightarrow{\mathcal{D}} P_X$. That is, the preimage of the transferred sample approximately conforms to $P_X$ under previous assumptions when $n$ is large.*

*Proof.* Since $\hat{\mathcal{P}}_\mathcal{E} \xrightarrow{\text{asymp.}} \mathcal{P}$, the proof immediately follows from (8). ☐

## 4 SAMPLE GENERATION USING THE KERNEL TRANSFER OPERATOR

At this point, the transferred sample $\Psi^*$, obtained by the kPF operator, remains an element of RKHS $\mathcal{G}$. To translate the samples back to the input space, we must find the preimage $x^*$ such that $\psi(x^*) = \Psi^*$.

### 4.1 Solving for an approximate preimage

Solving the preimage in kernel-based methods is known to be ill-posed (Mika et al., 1999) because the mapping $\psi(\cdot)$ is not necessarily surjective, i.e., a unique preimage $\mathbf{x}^* = \psi^{-1}(\Psi^*), \Psi^* \in \mathcal{H}$ may not exist. Often, an approximate preimage $\psi^{\dagger}(\mathbf{X}, \Psi^*) \approx \psi^{-1}(\Psi^*)$ is constructed instead based on relational properties among the training data in the RKHS. We consider two options in our framework **(1)** MDS-based method (Kwok & Tsang, 2004; Honeine & Richard, 2011),

$$\psi^{\dagger}_{\mathrm{MDS}}(\mathbf{X}, \Psi^*) = \tfrac{1}{2}(\mathbf{X}'\mathbf{X}'^{\top})^{-1}\mathbf{X}'(\mathrm{diag}(\mathbf{X}'^{\top}\mathbf{X}') - \mathbf{d}^{\top}), \text{ where } \forall i \in [\gamma], \mathbf{d}_i = \|l(\mathbf{x}'_i, \cdot) - \Psi^*\|_{\mathcal{G}} \quad (9)$$

which optimally preserves the distances in RKHS to the preimages in the input space, and **(2)** weighted Fréchet mean (Friedman et al., 2001), which in Euclidean space takes the form

$$\psi^{\dagger}_{\mathrm{wFM}}(\mathbf{X}, \Psi^*) = \psi^{\dagger}_{\mathrm{wFM}}(\mathbf{X}'; \mathbf{s}) = \mathbf{X}'\mathbf{s}/\|\mathbf{s}\|_1, \text{ where } \forall i \in [\gamma], \mathbf{s}_i = \langle l(\mathbf{x}'_i, \cdot), \Psi^* \rangle \quad (10)$$

where $\mathbf{X}'$ a neighborhood of $\gamma$ training samples based on pairwise distance or similarity in RKHS, following (Kwok & Tsang, 2004). The weighted Fréchet mean preimage uses the inner product weights $\langle \Psi^*, \psi(\mathbf{x}_i) \rangle$ as measures of similarities to interpolate training samples. On the toy data (as in Fig. 2), weighted Fréchet mean produces fewer samples that deviate from the true distribution and is easier to compute. Based on this observation, we use the weighted Fréchet mean as the preimage module for all experiments that requires samples, while acknowledging that other preimage methods can also be substituted in.

With all the ingredients in hand, we now present an algorithm for sample generation

---

**Algorithm 1** Sample Generation from kPF

1: **Input:** Training data $\mathbf{X} = \{\mathbf{x}_1, \ldots, \mathbf{x}_n\}$, Optional autoencoder $(E, D)$, input/output kernels $(k, l)$, neighborhood size $\gamma$
2: **Training**
3:     $\mathbf{X} = (E(\mathbf{x}_1), \ldots, E(\mathbf{x}_n))$ if $E$ is provided
4:     Sample $\{\mathbf{z}_i\}_{i \in [n]} \sim P_Z$ independently
5:     Construct $L, K \in R^{n \times n}$ s.t.
        $L_{ij} = l(\mathbf{x}_i, \mathbf{x}_j), K_{ij} = k(\mathbf{z}_i, \mathbf{z}_j)$
6:     $K_{\mathrm{inv}} = (K + \lambda nI)^{-1}$ or $K^{\dagger}$
7: **Inference**
8:     Generate new prior sample $\mathbf{z}^* \sim P_Z$
9:     $\mathbf{s} = L \cdot K_{\mathrm{inv}}[k(\mathbf{z}_1, \mathbf{z}^*) \ldots k(\mathbf{z}_n, \mathbf{z}^*)]^{\top}$
10:     $ind = \mathrm{argsort}(\mathbf{s})[-\gamma :]$
11:     $\mathbf{x}^* = \psi^{\dagger}_{\mathrm{wFM}}(\mathbf{X}[ind]; \mathbf{s}[ind])$.
12: **Output** $D(\mathbf{x}^*)$ if $D$ is provided else $\mathbf{x}^*$

---

using the kPF operator in Alg. 1. The idea is simple yet powerful: at training time, we construct the empirical kPF operator $\hat{\mathcal{P}}_{\mathcal{E}}$ using the training data $\{\mathbf{x}_i\}_{i \in [s]}$ and samples of the known prior $\{\mathbf{z}_i\}_{1 \in [n]}$. At test time, we will transfer new points sampled from $P_Z$ to feature maps in $\mathcal{H}$, and construct their preimages as the generated output samples.

### 4.2 Image generation

Image generation is a common application for generative models (Goodfellow et al., 2014; Dinh et al., 2017). While our proposal is not image specific, constructing sample preimages in a high dimensional space with limited training samples can be challenging, since the space of images is usually not dense in a reasonably sized neighborhood. However, empirically images often lie near a low dimensional manifold in the ambient space (Seung & Lee, 2000), and one may utilize an autoencoder (AE) $(E, D)$ to embed the images onto a latent space that represents coordinates on a learned manifold. If the learned manifold lies close to the true manifold, we can learn densities on the manifold directly (Dai & Wipf, 2019).

Therefore, for image generation tasks, the training data is first projected onto the latent space of a pretrained AE. Then, the operator will be constructed using the projected latent representations, and samples will be mapped back to image space with the decoder of AE. Our setup can be viewed analogously to other generative methods based on so called "*ex-post*" density estimation of latent variables (Ghosh et al., 2020). We also restrict the AE latent space to a hypersphere $\mathbf{S}^{n-1}$ to ensure that **(a)** $k(\cdot, \cdot)$ and $l(\cdot, \cdot)$ are bounded and **(b)** the space is geodesically convex and complete, which is required by the preimage computation. To compute the weighted Fréchet mean on a hypersphere, we adopt the recursive algorithm in Chakraborty & Vemuri (2015) (see appendix D for details).

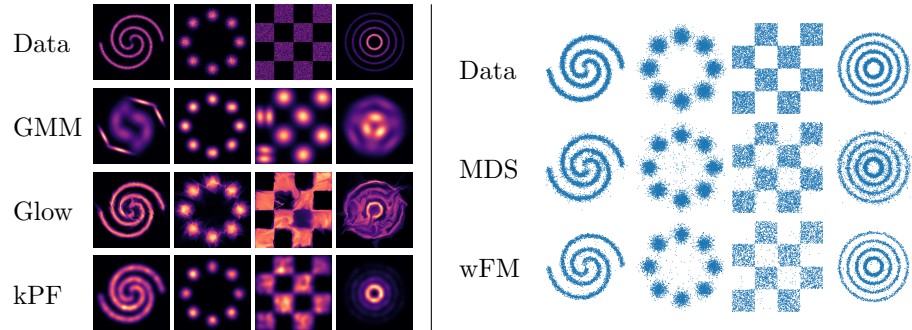

Figure 2: *Left figure:* Density estimation on 2D toy data. **Top to bottom: (1)** Training data samples, and learned densities of **(2)** GMM **(3)** Glow **(4)** Proposed kPF operator. More details in appendix. *Right figure:* Sample generation results. **Top:** Data samples **Middle:** MDS-based preimage samples **Bottom:** Weighted Fréchet mean samples.

## 5  EXPERIMENTAL RESULTS

**Goals.** In our experiments, we seek to answer three questions: **(a)** With sufficient data, can our method generate new data with comparable performance with other state-of-the-art generative models? **(b)** If only limited data samples were given, can our method still estimate the density with reasonable accuracy? **(c)** What are the runtime benefits, if any?

**Datasets/setup.** To answer the first question, we evaluate our method on standard vision datasets, including MNIST, CIFAR10, and CelebA, where the number of data samples is much larger than the latent dimension. We compare our results with other VAE variants (Two-stage VAE Dai & Wipf (2019), WAE Arjovsky et al. (2017), CV-VAE Ghosh et al. (2020)) and flow-based generative models (Glow Kingma & Dhariwal (2018), CAGlow Liu et al. (2019)) The second question is due to the broad use of kernel methods in small sample size settings. For this more challenging case, we randomly choose 100 training samples ($< 1\%$ of the full dataset) from CelebA and evaluate the quality of generation compared to other density approximation schemes. We also use a dataset of T1 Magnetic Resonance (MR) images from the Alzheimer's Disease Neuromaging Initiative (ADNI) study.

**Distribution transfer with many data samples.** We evaluate the quality by calculating the Fréchet Inception Distance (FID) (Heusel et al., 2017) with 10K generated images from each model. Here, we use a pretained regularized autoencoder (Ghosh et al., 2020) with a latent space restricted to the hypersphere (denoted by SRAE) to obtain *smooth* latent representations. We compare our kPF to competitive end-to-end deep generative baselines (i.e. flow and VAE variants) as well as other density estimation models over the same SRAE latent space. For the latent space models, we experimented with Glow (Kingma &

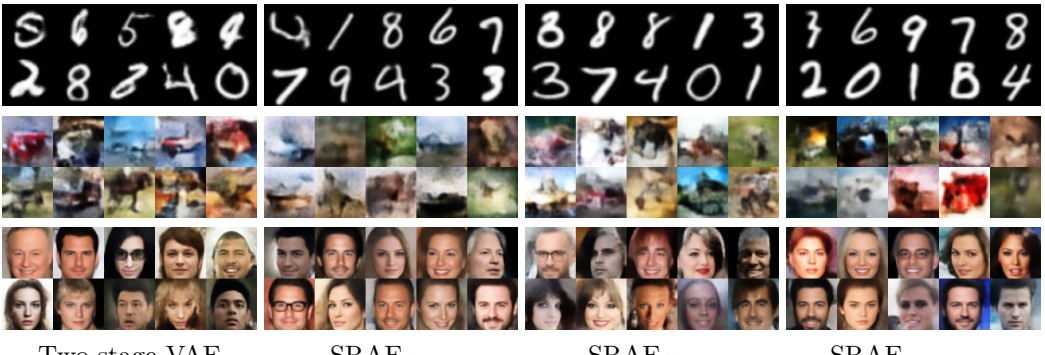

| Two-stage VAE | SRAE$_{Glow}$ | SRAE$_{GMM}$ | SRAE$_{NTK\text{-}kPF}$ |

Figure 3: Comparison of different sampling techniques using AE trained on CelebA 64x64. Left to right: samples of (1) Two-Stage VAE (Dai & Wipf, 2019) (2) SRAE$_{Glow}$ (Kingma & Dhariwal, 2018) (3) SRAE$_{GMM}$ (4) SRAE $_{NTK\text{-}kPF}$ using 10k latent points.

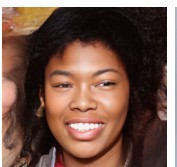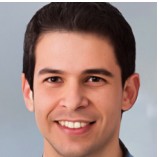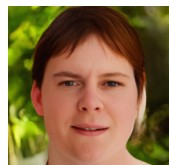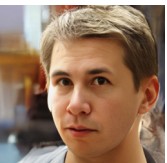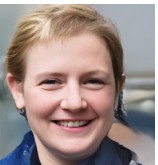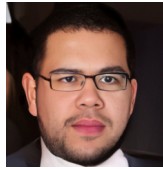

Figure 4: Representative samples from learned kPF on pre-trained NVAE latent space

Dhariwal, 2018), VAE, Gaussian mixture model (GMM), and two proposed kPF operators with Gaussian kernel (RBF-kPF) and NTK (NTK-kPF) as the input kernel. The use of NTK is motivated by promising results at the interface of kernel methods and neural networks (Jacot et al., 2018; Arora et al., 2020). Implementation details are included in the appendix.

Comparative results are shown in Table 2. We see that for images with structured feature spaces, e.g., MNIST and CelebA, our method matches other non-adversarial generative models, which provides evidence in support of the premise that the forward operator can be simplified. Further, we present qualitative results on all datasets (in Fig. 3), where we compare our kPF operator based model with other density estimation techniques on the latent space. Observe that our model generates comparable visual results as SRAE$_{\text{Glow}}$.

Since kPF learns the distribution on a pre-trained AE latent space for image generation, using a more powerful AE can offer improvements in generation quality. In Fig. 4, we present representative images by learning our kPF on NVAE Vahdat & Kautz (2020) latent space, pre-trained on the FFHQ dataset. NVAE builds a hierarchical prior and achieves state-of-the-art generation quality among VAEs. We see that kPF can indeed generate high-quality and diverse samples with the help of NVAE encoder/decoder. In fact, any AE/VAE may be substituted in, assuming that the latent space is *smooth*.

**Summary:** When a sufficient number of samples are available, our algorithm performs as well as the alternatives, which is attractive given the efficient training. In Fig. 5, we present comparative result of FIDs with respect to the training time. Since kPF can be computed in closed-form, it achieves significant training efficiency gain compared to other deep generative methods while delivering competitive generative quality.

**Distribution transfer with limited data samples.** Next, we present our evaluations when only a limited number of samples are available. Here, each of the density estimators was trained on latent representations of the same set of 100 randomly sampled CelebA images, and 10K images were generated to evaluate FID (see Table 3). Our method outperforms Glow and VAE, while offering competitive performance with GMM. Surprisingly, GMM remains a strong baseline for both tasks, which agrees with results in Ghosh et al. (2020). However, note that GMM is restricted by its parametric form and is less flexible than our method (as shown in Fig 2).

|  | MNIST | CIFAR | CelebA |
|---|---|---|---|
| Glow[‡] | 25.8 | - | 103.7 |
| CAGlow[‡] | 26.3 | - | 104.9 |
| Vanilla VAE | 36.5 | 111.0 | 52.1 |
| CV-VAE[†] | 33.8 | 94.8 | 48.9 |
| WAE[†] | 20.4 | 117.4 | 53.7 |
| Two-stage VAE | 16.5 | 110.3 | 44.7 |
| SRAE$_{\text{Glow}}$ | **15.5** | 85.9 | **35.0** |
| SRAE$_{\text{VAE}}$ | 17.2 | 198.0 | 48.9 |
| SRAE$_{\text{GMM}}$ | 16.7 | 79.2 | 42.0 |
| SRAE$_{\text{RBF-kPF}}$(*ours*) | 19.7 | 77.9 | 41.9 |
| SRAE$_{\text{NTK-kPF}}$(*ours*) | 19.5 | **77.5** | 41.0 |

Table 2: Comparative FID values. SRAE indicates an autoencoder with hyperspherical latent space and spectral regularization following Ghosh et al. (2020). Results reported from ‡: Liu et al. (2019). †: Ghosh et al. (2020).

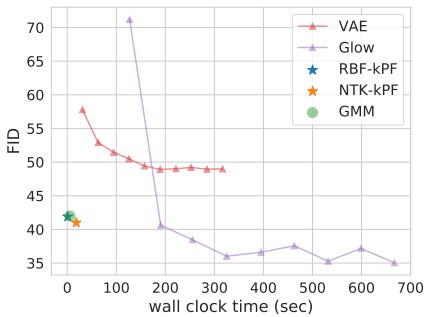

Figure 5: FID *versus* training time for latent space models. All models are learned on the latent representations encoded by the same pre-trained SRAE.

Learning from few samples is common in biomedical applications where acquisition is costly.

| VAE | Glow | GMM | RBF-kPF | NTK-kPF |
|------|------|------|---------|---------|
| 59.3 | 77.0 | 39.6 | 40.6 | 40.9 |

Table 3: FID values for few samples setting density approximation on CelebA.

Motivated by interest in making synthetic but statistic preserving data (rather than the real patient records) publicly available to researchers (see NIH N3C Data Overview), we present results on generating high-resolution ($160 \times 196 \times 160$) brain images: 183 samples from group AD (diagnosed as Alzheimer's disease) and 291 samples from group CN (control normals). For $n = 474 \ll d = 5017600$, using our kernel operator, we can generate high-quality samples that are in-distribution. We present comparative results with VAEs. The generated samples in Fig. 6 clearly show that our method generates sharper images. To check if the results are also scientifically meaningful, we test consistency between group difference testing (i.e., cases versus controls differential tests on each voxel) on the real images (groups were AD and CN) and the same test was performed on the generated samples (AD and CN groups), using a FWER corrected two-sample $t$-test Ashburner & Friston (2000). The results (see Fig 6) show that while there is a deterioration in regions identified to be affected by disease (different across groups), many statistically-significant regions from tests on the real images are preserved in voxel-wise tests on the generated images.

**Summary:** We achieve improvements in the small sample size setting compared to other generative methods. This is useful in many data-poor settings. For larger datasets, our method still compares competitively with alternatives, but with a smaller resource footprint.

## 6 LIMITATIONS

Our proposed simplifications can be variously useful, but deriving the density of the posterior given a mean embedding or providing an exact preimage for the generated sample in RKHS is unresolved at this time. While density estimation from kPF has been partially addressed in Schuster et al. (2020b), finding the pre-image is often ill-posed. The weighted Fréchet mean preimage only provides an approximate solution and evaluated empirically, and due to the interpolation-based sampling strategy, samples cannot be obtained beyond the convex hull of training examples. Making $Z$ and $X$ independent RVs also limits its use for certain downstream task such as representation learning or semantic clustering. Finally, like other kernel-based methods, the sup-quadratic memory/compute cost can be a bottleneck on large datasets and kernel approximation (e.g. (Rahimi & Recht, 2009)) may have to be applied; we discuss this in appendix F.

## 7 CONCLUSIONS

We show that using recent developments in regularized autoencoders, a linear kernel transfer operator can potentially be an efficient substitute for the forward operator in some generative models, if some compromise in capabilities/performance is acceptable. Our proposal, despite its simplicity, shows comparable empirical results to other generative models, while offering efficiency benefits. Results on brain images also show promise for applications to high-resolution 3D imaging data generation, which is being pursued actively in the community.

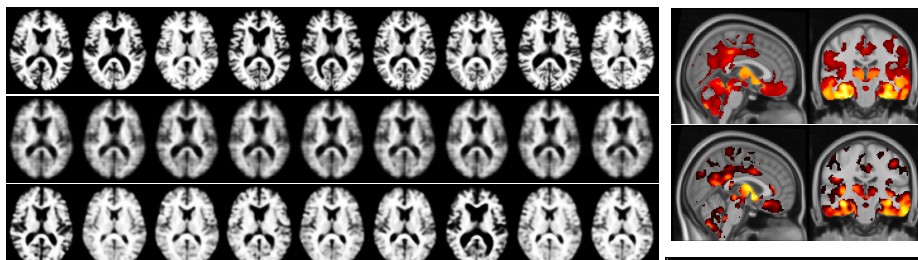

Figure 6: **Left.** *Top: data*, generated samples of *Middle: VAE , Bottom: kPF with SRAE.* **Right.** Statistically significant regions *Top: voxel-wise tests on real data, Bottom: voxel-wise tests on generated samples* are shown in negative log $p$-value thresholded at $\alpha = 0.01$.

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

## A    Choice of Kernel is relevant yet flexible

In some cases, one would focus on identifying (finite) eigenfuntions and modes of the underlying operator (Williams et al., 2015b; Brunton et al., 2016). But rather than finding certain modes that best characterize the dynamics, we care most about minimizing the error of the transferred density and therefore whether the span of functions is rich/expressive enough. In particular, condition (iii) in Proposition 3.2 requires an input RKHS spanned by sufficiently rich bases (Fukumizu et al., 2013). For this reason, the choice of kernel here cannot be completely ignored since it determines the family of functions contained in the induced RKHS.

To explore the appropriate kernel setup for our application, we empirically evaluate the effect of using several different kernels via a simple experiment on MNIST. We first train an autoencoder to embed MNIST digits on to a hypersphere $S^2$, then generate samples from kPF by the procedure described by Alg. 1 using the respective kernel function as the input kernel $k$. Subplot (b) and (c) in Fig. 7 show the generated samples using Radial Basis Function (RBF) kernel and arc-cosine kernel, respectively. Observe that the choice of kernel has an influence on the sample population, and a kernel function with superior empirical behavior is desirable.

Motivated by this observation, we evaluated the Neural Tangent Kernel (NTK) (Jacot et al., 2018), a well-studied neural kernel in recent works. We use it for a few reasons, **(a)** NTK, in theory, corresponds to a trained infinitely-wide neural network, which spans a rich set of functions that satisfies the assumption. **(b)** For well-conditioned inputs (i.e., no duplicates) on hypersphere, the positive-definiteness of NTK is proved in (Jacot et al., 2018). Therefore, invertibility of the Gram matrix $K_{ZZ} = \Phi^T \Phi$ is *almost guaranteed* if the prior distribution $p_Z$ is restricted on a hypersphere **(c)** NTK can be non-asymptotically approximated (Arora et al., 2019). **(d)** Unlike other parametric kernels such as RBF kernels, NTK is less sensitive to hyperparameters, as long as the number of units used is large enough (Arora et al., 2019). Subplot (d) of Fig. 7 shows that kPF learned with NTK as input kernel is able to generate samples that are more consistent with the data distribution. However, we should note that NTK is merely a convenient choice of kernel that requires less tuning, and is not otherwise central to our work. In fact, as shown in our experiment in Tab. 2, a well-tuned RBF kernel may also achieve a similar performance. Indeed, in practice, any suitable choice of kernel may be conveniently adopted into the proposed framework without major modifications.

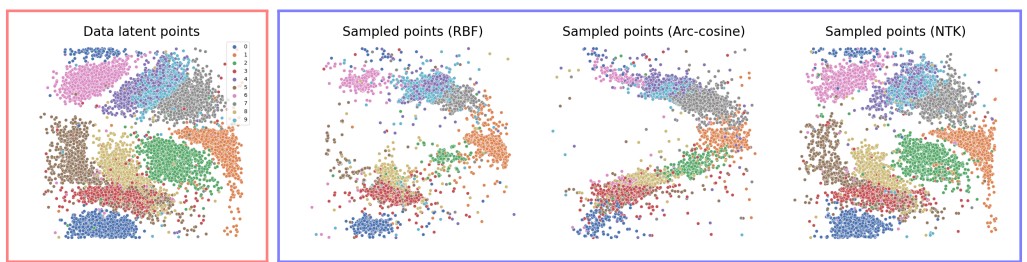

Figure 7: 10k samples from MNIST dataset (*left to right*) (a) projected on $\mathbf{S}^2$ shown in $(\theta, \phi)$ using auto-encoder, and 10K generated samples from kPF with input kernel of type (b) RBF (c) arccos (d) NTK. Color of sampled points represents the class of their nearest training set neighbor in the output RKHS.

## B    DENSITY ESTIMATION WITH KPF OPERATOR

The displayed transferred density with the kPF operator on toy data in Fig. 1 is *approximated* using the empirical kernel conditional density operator (CDO) (Schuster et al., 2020a), since there is currently no known methods that can exactly reconstruct density from the transferred mean embeddings. The marginalized transferred density $p_{\mathcal{P}_{\mathcal{E}}}$ has the following form

$$p_{\mathcal{P}_{\mathcal{E}}} = C_{\rho}^{-1}\mathcal{P}_{\mathcal{E}}\mu_Z = C_{\rho}^{-1}C_{XZ}C_{ZZ}^{-1}\mu_Z, \tag{11}$$

where $C_{\rho} = E_{y\sim\rho}[l(y,\cdot)\otimes l(y,\cdot)]$ is the covariance operator of a independent reference density $\rho$ in $\mathcal{G}$. The above density function is also an element of RKHS $\mathcal{G}$, and therefore we can evaluate the density at $x^*$ by using the reproducing property $p_{\mathcal{P}_{\mathcal{E}}}(x^*) = \langle p_{\mathcal{P}_{\mathcal{E}}}, l(x^*,\cdot)\rangle$. The results in Schuster et al. (2020a) show that the empirical estimate of $p_{\mathcal{P}_{\mathcal{E}}}$ may be constructed from $m$ samples of the reference density $\{y_i\}_{i\in[m]}$ and $n$ training samples $\{x_i\}_{i\in[n]}$ and $\{z_i\}_{i\in[n]}$ as

$$\hat{p}_{\mathcal{P}_{\mathcal{E}}} = (\hat{C}_{\rho} + \alpha'I)^{-1}\hat{C}_{XZ}(\hat{C}_{ZZ} + \alpha I)^{-1}\hat{\mu}_Z = \sum_{i=1}^{m}\beta_i l(y_i,\cdot) \tag{12}$$

where

$$\beta = m^{-2}(L_Y + \alpha'I)^{-2}L_{YX}(K_Z + \alpha I)^{-1}\Phi^{\top}\hat{\mu}_Z \tag{13}$$

and

$$L_Y = [l(y_i,y_j)]_{ij} \in \mathbb{R}^{m\times m}, L_{YX} = [l(y_i,x_j)]_{ij} \in \mathbb{R}^{m\times n}, K_Z = [k(z_i,z_j)]_{ij} \in \mathbb{R}^{n\times n} \tag{14}$$

In Fig. 8, we use a uniform density in the square $(\pm 4, \pm 4)$ as the reference density $\rho$ and constructed $\hat{p}_{\mathcal{P}_{\mathcal{E}}}$ using $m = 10000$ samples $\{y_i\}_{i\in[m]}$ from $\rho$. Due to the form of the empirical kernel CDO, where the estimated density function $\hat{p}_{\mathcal{P}_{\mathcal{E}}}$ is a linear combination of $\{l(y_i,\cdot)\}_{i\in[m]}$ (as in Eq. 12), the approximation can be inaccurate if reference samples are relatively sparse and the densities are 'sharp'. In those cases, to obtain a better density estimate, we may either increase the number of reference samples used to construct the empirical CDO (which can be computationally difficult due to the need to compute $(L_Y + \alpha'I)^{-2}$), or, with some prior knowledge to the true distribution, choose a reference density which is localized around the ground truth density.

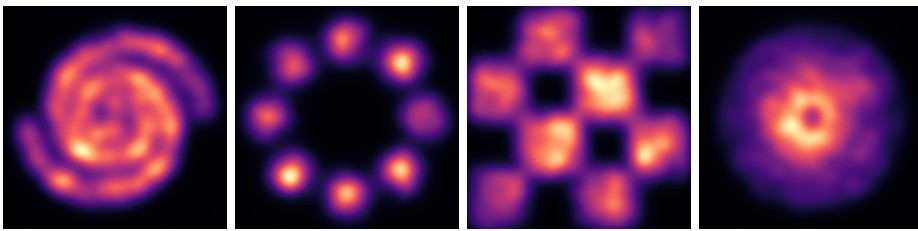

Figure 8: Inaccurate density estimation result from kernel CDO using 10k samples from uniform reference density $\rho$

Therefore, to show a more faithful density estimate of the transferred distribution for visualization purpose, we use a composite of the uniform density and the true density with weight $4:1$ as the reference density $\rho$ in Fig. 2. In this case, approximately 20% of the reference samples concentrates around the high-density areas of the true density, which helps to form a better basis for $\hat{p}_{\mathcal{P}_{\mathcal{E}}}$. Note that the choice of $\rho$ does not affect the transferred density embedding $\hat{\mathcal{P}}_{\mathcal{E}}\hat{\mu}_Z$ since it is independent of $p_X$ and $p_Z$. After this modification, the reconstructed density more accurately reflects the true density compared to GMM and GLOW, indicating the transferred distribution by kPF in RKHS matches better to the ground truth distribution. This is also reflected in the generated samples in Fig. 9, where samples generated by the kPF operator are clearly more aligned with the ground truth distribution.

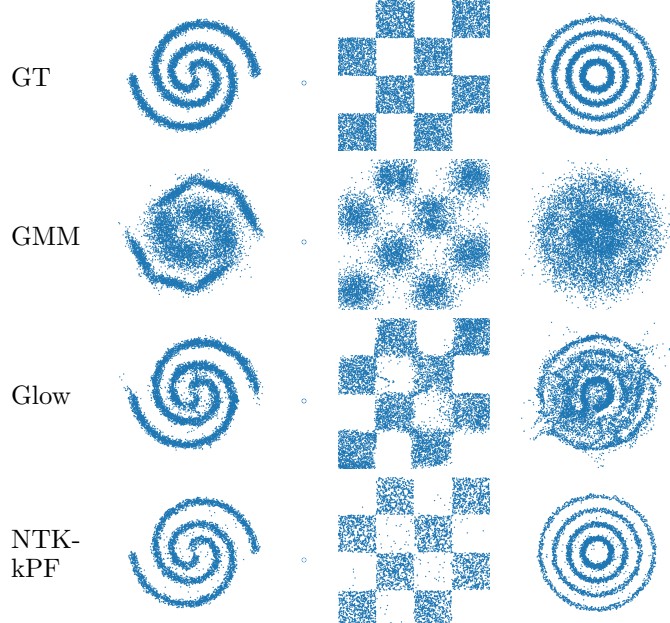

Figure 9: Sample comparisons of the distribution matching methods

## C  EFFECT OF $\gamma$ ON SAMPLE QUALITY

In the sampling stage, our proposed method finds the approximate preimage of the transferred kernel embeddings by taking the weighted Fréchet mean of the top $\gamma$ neighbors among the training samples. The choice of $\gamma$ therefore influences the quality of generation.

From Figure 10, we can observe that, in general, FID worsens as $\gamma$ increases. This observation aligns with our intuition of preserving only the local similarities represented by the kernel, and similar ideas have been previously used in the literature (Hastie et al., 2001; Kwok & Tsang, 2004). However, significantly decreasing $\gamma$ leads to the undesirable result where the generator merely generates the training samples (in the extreme case where $\gamma = 1$, generated samples will just be reconstructions of training samples). Therefore, in our experiments, we choose $\gamma = 5$ to achieve a balance between generation quality and the distance to training samples.

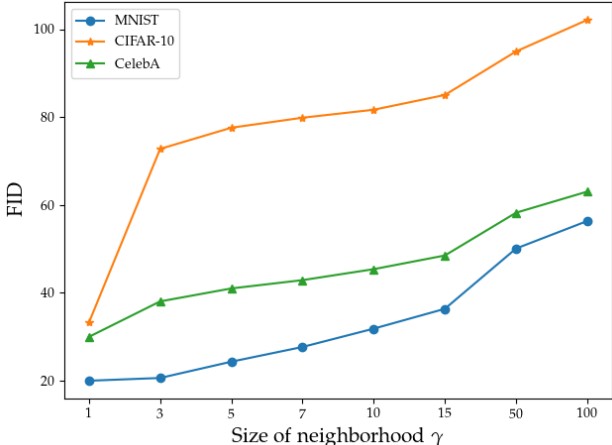

Figure 10: FID *versus* $\gamma$ on a few computer vision datasets.

# D WEIGHTED FRÉCHET MEAN ON THE HYPERSPHERE

While the weighted Fréchet Mean in Euclidean space can be computed in closed-form as a weighted arithmetic mean (as in Eq. 10), on the hypersphere there is no known closed-form solution. Thus, we adopt the iterative algorithm in (Chakraborty & Vemuri, 2015) for an approximate solution given data points $\mathbf{X} = \{\mathbf{x}_1 \dots \mathbf{x}_\gamma\}$ and weight vector $\mathbf{s}$:

$$M_1 = \mathbf{x}_1$$
$$M_{i+1} := \cos(\|\mathbf{s}_{i+1}\mathbf{v}\|)M_i + \sin(\|\mathbf{s}_{i+1}\mathbf{v}\|)\frac{\mathbf{v}}{\|\mathbf{v}\|}$$

where, $\mathbf{v} = \frac{\theta}{\sin(\theta)}(X_{i+1} - M_i \cos(\theta))$, $\theta = \arccos(M_i^t X_{i+1})$. This algorithm iterates through the data points once, yielding a complexity of only $O(\gamma d)$, where $d$ is the dimension of $\mathcal{X}$. Under the prescribed iteration, $M_n$ converges asymptotically to the true Fréchet mean for finite data points. We refer the readers to (Chakraborty & Vemuri, 2015) for further details.

## E  Fast approximation of Moore-Penrose inverse

When computing the inverted kernel matrix $K_{\text{inv}}$ in Algo. 1, conventional approaches typically performs SVD or Cholesky decomposition. Both procedures are hard to parallelize, and therefore, can be slow when $K$ is large. Alternatively, we can utilize an iterative procedure proposed in Razavi et al. (2014) to approximate the Moore-Penrose inverse.

$$Z_1 = K/(\|K\|_1 \|K\|_\infty) \tag{15}$$
$$Z_{i+1} := Z_i(13I - KZ_i(15I - KZ_i(7I - KZ_i))) \tag{16}$$

where

$$\|K\|_1 = \max_j \sum_{i=0}^{n} K_{ij}, \|K\|_\infty = \max_i \sum_{j=0}^{n} K_{ij} \tag{17}$$

Since this iterative procedure mostly involves matrix multiplications, it can be efficiently parallelized and implemented on GPU. The same procedure has also seen success in approximating large self-attention matrices in language modeling (Xiong et al., 2021). For the NVAE experiment, we run this iteration for 10 steps and use $K_{\text{inv}} = Z_{10}$.

## F   NYSTROM APPROXIMATION OF KPF

Due to the need to store and compute a kernel matrix inverse $(K + \lambda nI)^{-1}$ or $K^{\dagger}$, the memory and computational complexity of kPF is at least $O(n^2)$ and $O(n^3)$, respectively. The sup-quadratic complexity hinders the use of kPF on extremely large datasets. In our experiments, we already adopted a simple subsampling strategy which randomly select 10k training samples from each dataset ($\leq$ 50k samples) to fit our hardware configuration which works well. But for larger datasets with potentially more modes, a larger set of subsamples must be considered, and in those cases kPF may not be suitable for commodity/affordable hardware. In order to overcome this problem, we can combine kPF with conventional kernel approximation methods such as the Nyström method (Williams & Seeger, 2001).

Let $(\mathbf{X}_{\star}, \mathbf{Z}_{\star})$ be a size $v$ subset of the training set (which we refer to as the *landmark points*) and $(\Psi_{\star}, \Phi_{\star})$ be their corresponding kernel feature maps. The weighting coefficients $\mathbf{s}$ for each prior sample $z^* \sim Z$ derived in Alg. 1 can be approximated by

$$
\begin{aligned}
\mathbf{s} &= L(K + \lambda nI)^{\dagger} \Phi^{\top} k(z^{\star}, \cdot) \\
&\approx L_{\Psi_{\star}} W_{\Psi_{\star}}^{\dagger} L_{\Psi_{\star}}^{\top} (K_{\Phi_{\star}} W_{\Phi_{\star}}^{\dagger} K_{\Phi_{\star}}^{\top} + \lambda nI)^{\dagger} \Phi^{\top} k(z^{\star}, \cdot) \\
&= L_{\Psi_{\star}} W_{\Psi_{\star}}^{\dagger} L_{\Psi_{\star}}^{\top} (\lambda n)^{-1} (I - K_{\Phi_{\star}}^{\top} (\lambda n W_{\Phi_{\star}}^{\dagger} + K_{\Phi_{\star}} K_{\Phi_{\star}}^{\top})^{\dagger} K_{\Phi_{\star}}) \Phi^{\top} k(z^{\star}, \cdot) \quad (18)
\end{aligned}
$$

where $L_{\Psi_{\star}} = \Psi^{\top} \Psi_{\star} \in \mathbb{R}^{n \times v}$, $W_{\Psi_{\star}} = \Psi_{\star}^{\top} \Psi_{\star} \in \mathbb{R}^{v \times v}$, $K_{\Phi_{\star}} = \Phi^{\top} \Phi_{\star} \in \mathbb{R}^{n \times v}$, $W_{\Phi_{\star}} = \Phi_{\star}^{\top} \Phi_{\star} \in \mathbb{R}^{v \times v}$, and the last identity is due to applying the Woodbury formula on $(K_{\Phi_{\star}} W_{\Phi_{\star}}^{\dagger} K_{\Phi_{\star}}^{\top} + \lambda nI)^{\dagger}$. Assuming $v \ll n$, the memory complexity is reduced to $O(nv)$ and the computation complexity to $O(nv^2)$.

We empirically evaluated the Nyström-approximated kPF on the CelebA experiment and present the result in Tab. 4. It can be observed that when $v$ is sufficiently large, the performance of Nyström approximated kPFs is as good as the ones using the full kernel matrices.

| $n$ \ $v$ | 100 | 500 | 1000 | w/o Approximation |
|---|---|---|---|---|
| 10,000 | 45.9 | 41.6 | 42.3 | 41.8 |
| 30,000 | 46.3 | 40.5 | 42.4 | - |
| 50,000 | 45.2 | 44.1 | 42.0 | - |

Table 4: FIDs of samples generated by Nyström-approximated NTK-kPF on CelebA. $n$ denotes the size of the training data subset we consider in computing kPF, while $v$ denotes the size of selected landmark points for Nyström approximation. Without approximation, we cannot fit the kernel matrices onto a GPU with 11GB RAM when $n > 10,000$. It is worth noting that the approximated kPFs can perform similarly to the full kPF even with $v < 0.05n$, which indicates that Nyström approximation does not sacrifice much in terms of performance while delivering significant efficiency gain.

## G    Does kPF Memorize Training Data?

Since in kPF, samples are generated by linearly interpolating between training samples, it is natural to wonder whether it 'fools' the metrics by simply replicating training samples. For comparison, we consider an alternative scheme that generates data through direct manipulation of the training data, namely Kernel Density Estimation (KDE).

We fit KDEs by varying noise levels $\sigma$ and compare their FIDs and nearest samples in the latent space to kPF in Fig. 11. We observe that, although KDE can reach very low FIDs when $\sigma$ is small, almost all new samples closely resemble some instance in the training set, which is a clear indication of memorization. In contrast, kPF can generate diverse samples that do not simply replicate the observed data.

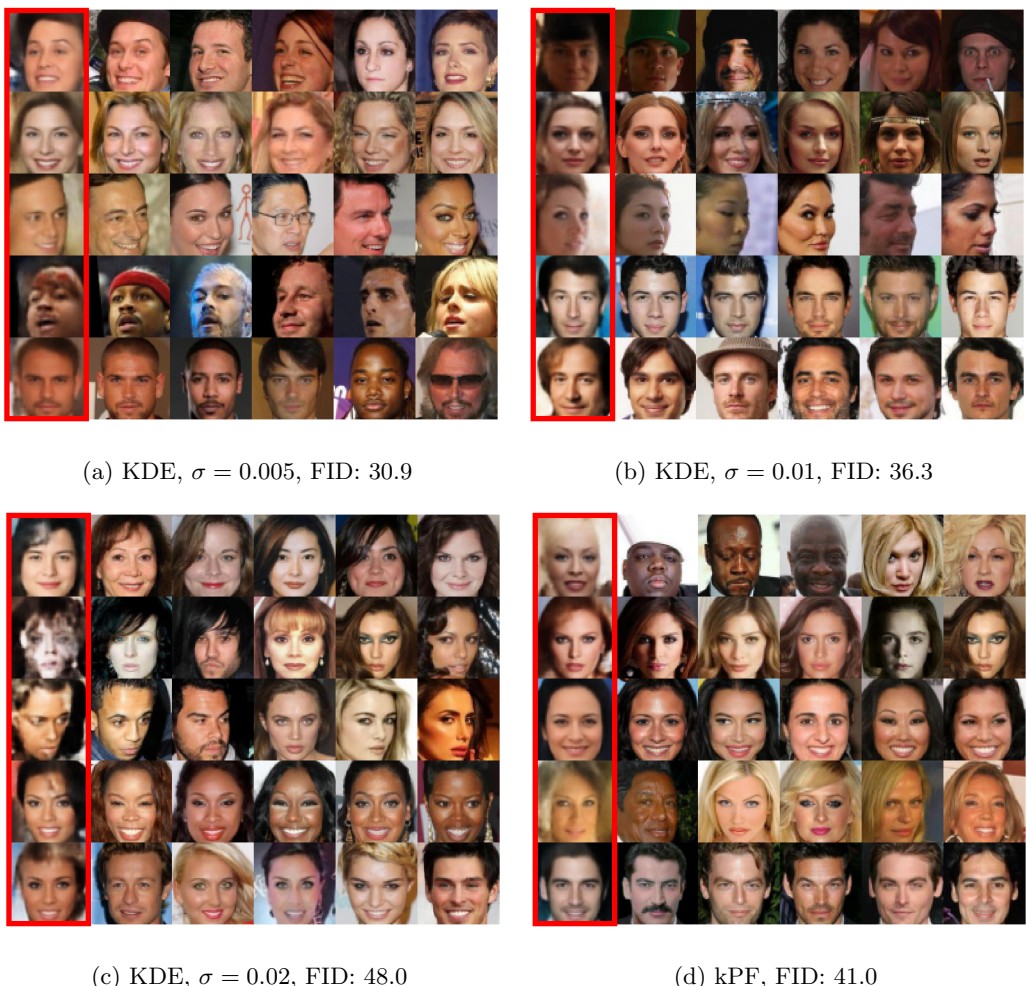

(a) KDE, $\sigma = 0.005$, FID: 30.9

(b) KDE, $\sigma = 0.01$, FID: 36.3

(c) KDE, $\sigma = 0.02$, FID: 48.0

(d) kPF, FID: 41.0

Figure 11: Comparing KDE to kPF. Generated samples are presented in ☐, followed by their 5 nearest neighbors in the latent space (ordered from closest to furthest)

# H    ASSESSING SAMPLE DIVERSITY

Although FID is one of the most frequently used measures for assessing sample quality of generative models, certain diversity considerations, such as mode collapse, may not be conveniently deduced from it (Sajjadi et al., 2018). To enable explicit examination of generative models with respect to both accuracy (i.e., generating samples within the support of the data distribution) and diversity (i.e., covering the support of the data distribution as much as possible), Sajjadi et al. (2018) proposed an approach to evaluate generative models with generalized definitions of *precision* and *recall* between distributions. Quality of generation can then be assessed by evaluating the PRD curve, which depicts the trade-offs between accuracy (precision) and diversity (recall). We present the PRD curves in Fig. 12. The observations align with our results in Tab. 2 and kPF performs competitively in both accuracy and sample diversity.

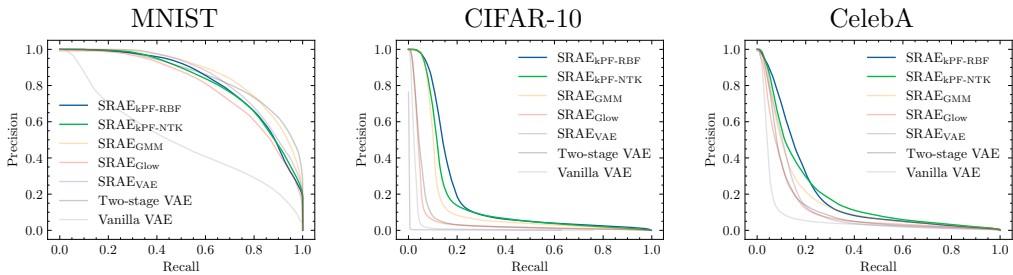

Figure 12: PRD curves on all datasets. kPF is competitive to the other methods in terms of Area Under Curve (AUC)

# I EXPLORING KERNEL CONFIGURATIONS

To investigate the implication of kernel choices on generation quality, we tested 25 different kernel configurations for CelebA generation (results are presented in Tab. 5). For RBF kernels used in the CelebA experiments of the main text, we use a bandwidth of $\sigma_{in} = \sqrt{2|\mathcal{Z}|}/8 \approx 0.71$ when used as input kernel and $\sigma_{out} \approx 0.34$, and we adopt the same notation here.

| Input kernel \ Output kernel | RBF $(\sigma = \sigma_{out}/4)$ | RBF $(\sigma = \sigma_{out}/2)$ | RBF $(\sigma = \sigma_{out})$ | RBF $(\sigma = 2\sigma_{out})$ | RBF $(\sigma = 4\sigma_{out})$ |
|---|---|---|---|---|---|
| RBF $(\sigma = \sigma_{in}/2)$ | 41.50 | 41.20 | 41.21 | 50.92 | 66.06 |
| RBF $(\sigma = \sigma_{in})$ | 41.90 | 42.11 | 41.91 | 45.83 | 50.70 |
| RBF $(\sigma = 2\sigma_{in})$ | 42.20 | 42.82 | 42.69 | 65.76 | 70.19 |
| NTK $(L = 8, w = 10,000)$ | 41.90 | 41.56 | 41.73 | **37.86** | 37.89 |
| Arccos $(L = 1, \deg = 1)$ | 41.71 | 42.03 | 42.22 | 52.83 | 63.13 |

Table 5: FID table for different kernel configurations.

It can be seen that kernel configurations and parameters indeed has a non-trivial impact on the generation quality, with NTK-kPF being the most robust to the choice of parameters. This aligns with our previous observations and offers some support for using NTK as an input kernel despite the additional compute cost.

## J  Experimental Details

In this section, we provide the detailed specifications for all of our experiments. We have also provided our code in the supplemental material.

### J.1  Density Estimation on Toy Densities

We generated 10000 samples from each of the toy densities to learn the kPF operator. The input kernel $k$ is a ReLU-activated NTK corresponding to a fully-connected network with depth $L = 4$ and width $w = 10000$ at each layer, and the output kernel $l$ is a Gaussian kernel. Unless specified otherwise, we always uses a Gaussian kernel as the output kernel for the remainder of this appendix. The bandwidth of the output kernel was adjusted separately for density estimation and sampling for the purpose of demonstration. For comparison, we also fit/estimate a 10-component GMM and a Glow model with 50 coupling layers, where each of them were trained until convergence.

### J.2  Image Generation with Computer Vision Datasets

To generate results in Tab. 2 and Tab. 3, we first trained an autoencoder for each dataset following the model setup in (Ghosh et al., 2020), which uses a modified Wasserstein autoencoder (Tolstikhin et al., 2018) architecture with batch normalization. Additionally, we applied spectral normalization on both the encoder and the decoder, following (Ghosh et al., 2020), to obtain a regularized autoencoder. The latent representations were projected onto a hypersphere before decoding to image space. We trained the models on two NVIDIA GTX 1080TI GPUs. A detailed model specification is provided below in Table 6.

We used an NTK with $L = 8$ and $w = 10000$ as the input kernel $k$ (i.e. the embedding kernel of $p_Z$) for NTK-kPF, and a Gaussian kernel with bandwidth $\sigma_{in} = \sqrt{2|\mathcal{Z}|}/8$ for RBF-kPF. The bandwidth for the output Gaussian kernel is selected by grid search over $\{2^{-i} * \sigma_{data}|i \in [8]\}$, where $\sigma_{data}$ is the empirical data standard deviation, based on cross-validation of a degree 3 polynomial kernel MMD between the sampled and the ground-truth latent points. Further, to mitigate the deterioration of performance of kernel methods in a high-dimensional setting due to the curse of dimensionality (Evangelista et al., 2006), in practice, we model $\mathcal{Z}$ as a space with fewer dimensions than the input space $\mathcal{X}$. As a rule of the thumb, we choose $\mathcal{Z}$ such that $|\mathcal{Z}| = |\mathcal{X}|/4$.

To generate images from kPF learned on NVAE latent space, we used the pre-trained checkpoints provided in (Vahdat & Kautz, 2020) to obtain the latent embeddings for 2000 FFHQ images. We then construct the kPF from the concatenated latent space of the lowest resolution ($8 \times 8$). During sampling, prior samples at those resolutions are replace by the kPF samples, while for other resolutions samples remain generated from inferred Gaussian distributions. The batchnorm statistics were readjusted for 500 iterations following (Vahdat & Kautz, 2020). We use rbf kernels as input and output kernels, with bandwidths $\sigma_k$, $\sigma_l$ chosen by the *median heuristic* ($\sim 100$ for input and $\sim 70$ for output in our experiments).

### J.3  Image Generation for Brain Images

For the high-resolution brain imaging dataset, we used a custom version of ResNet (He et al., 2016) with 3D convolutions. The detailed architecture is shown in Fig. 7. Due to the large size of the data, we trained the model on 4 NVIDIA Tesla V100 GPUs.

**Mandatory ADNI statement regarding data use.**  Data used in preparation of this article were obtained from the Alzheimer's Disease Neuroimaging Initiative (ADNI) database (adni.loni.usc.edu). As such, the investigators within the ADNI contributed to the design and implementation of ADNI and/or provided data but did not participate in analysis or writing of this report. A complete listing of ADNI investigators can be found in the ADNI Acknowledgement List.

The T1 MR brain dataset we utilize consists of images from 184 subjects diagnosed with Alzheimers's disease and 292 healthy controls/ normal subjects. Images were first coregistered

| | MNIST | CIFAR-10 | CelebA |
|---|---|---|---|
| Encoder | $\text{Conv}_{128}^{4\times4}$ $\text{Conv}_{256}^{4\times4}$ $\text{Conv}_{512}^{4\times4}$ $\text{Conv}_{1024}^{4\times4}$ | $\text{Conv}_{128}^{4\times4}$ $\text{Conv}_{256}^{4\times4}$ $\text{Conv}_{512}^{4\times4}$ $\text{Conv}_{1024}^{4\times4}$ | $\text{Conv}_{128}^{5\times5}$ $\text{Conv}_{256}^{5\times5}$ $\text{Conv}_{512}^{5\times5}$ $\text{Conv}_{1024}^{5\times5}$ |
| Decoder | $\text{ConvT}_{512}^{4\times4}$ $\text{ConvT}_{256}^{4\times4}$ $\text{ConvT}_{1}^{4\times4}$ | $\text{ConvT}_{512}^{4\times4}$ $\text{ConvT}_{256}^{4\times4}$ $\text{ConvT}_{3}^{4\times4}$ | $\text{ConvT}_{512}^{5\times5}$ $\text{ConvT}_{256}^{5\times5}$ $\text{ConvT}_{128}^{5\times5}$ $\text{ConvT}_{3}^{5\times5}$ |

Table 6: Model architecture for computer vision experiments. Subscript denotes the number of output channels and superscript denotes the window size of the convolution kernel. Batch normalization and activation is applied between each pair of convolution layers

| | |
|---|---|
| Encoder | 5x5 conv, stride 4 |
| | $\text{ResBlock}_{64} \times 2$ |
| | $\text{ResBlock}_{64} \times 2$ |
| | $\text{ResBlock}_{128} \times 2$ |
| | $\text{ResBlock}_{256} \times 2$ |
| Decoder | $\text{ResBlock}_{256} \times 2$ |
| | $\text{ResBlock}_{128} \times 2$ |
| | $\text{ResBlock}_{64} \times 2$ |
| | $\text{ResBlock}_{64} \times 2$ |
| | 5x5 conv, stride 4 |

Table 7: Model architecture for experiments for image generation on brain imaging dataset. Subscript denotes the number of output channels. Upsampling and downsampling are performed using strided convolutions.

to a MNI template and segmented to preserve only the white matter and grey matter. Then, all images were resliced and resized to $160 \times 196 \times 160$ and rescaled to the range of $[-1, 1]$. Voxel-based morphometry (VBM) was used to obtain the $p$-value map of data and generated images.

# K    MORE SAMPLES

In this section we present additional uncurated set of samples on MNIST, CIFAR-10, CelebA based on pre-trained SRAE and FFHQ based on NVAE. From the figures, it can be seen that kPF produces consistent and diverse samples, often better in quality than the alternatives.

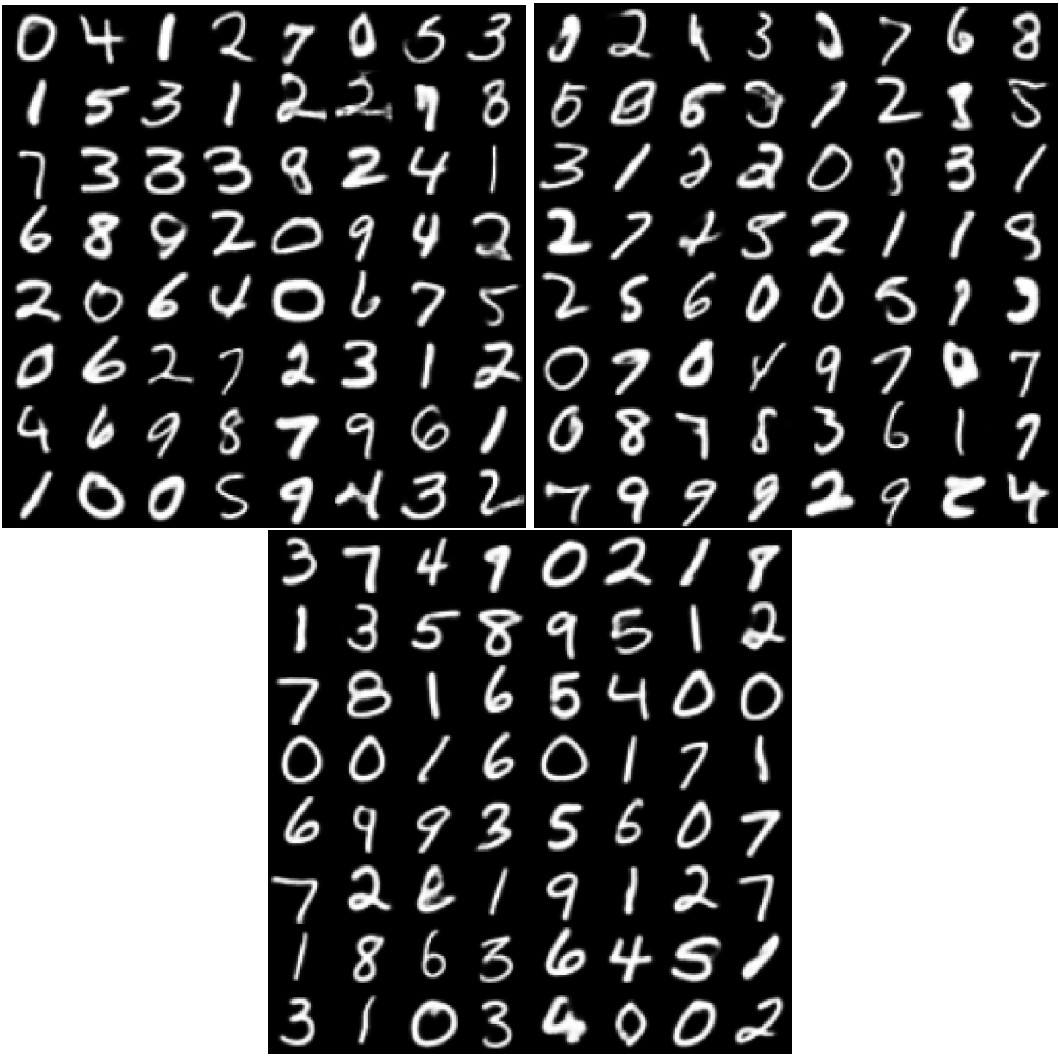

Figure 13: MNIST results from SRAE$_{\text{GMM}}$ (top left), SRAE$_{Glow}$ (top right) and our SRAE$_{\text{NTK-kPF}}$.

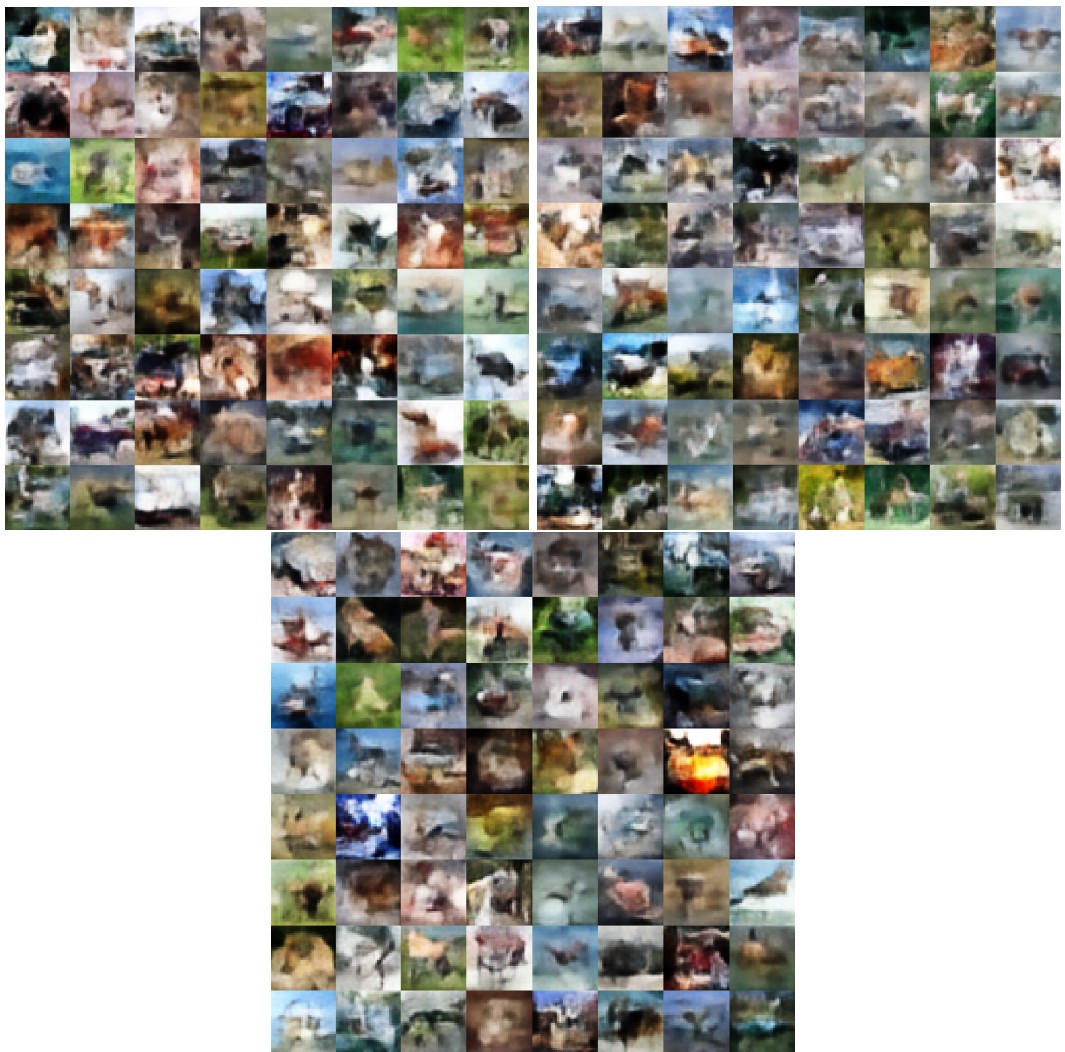

Figure 14: CIFAR results from SRAE$_{\text{GMM}}$ (top left), SRAE$_{Glow}$ (top right) and our SRAE$_{\text{NTK-kPF}}$.

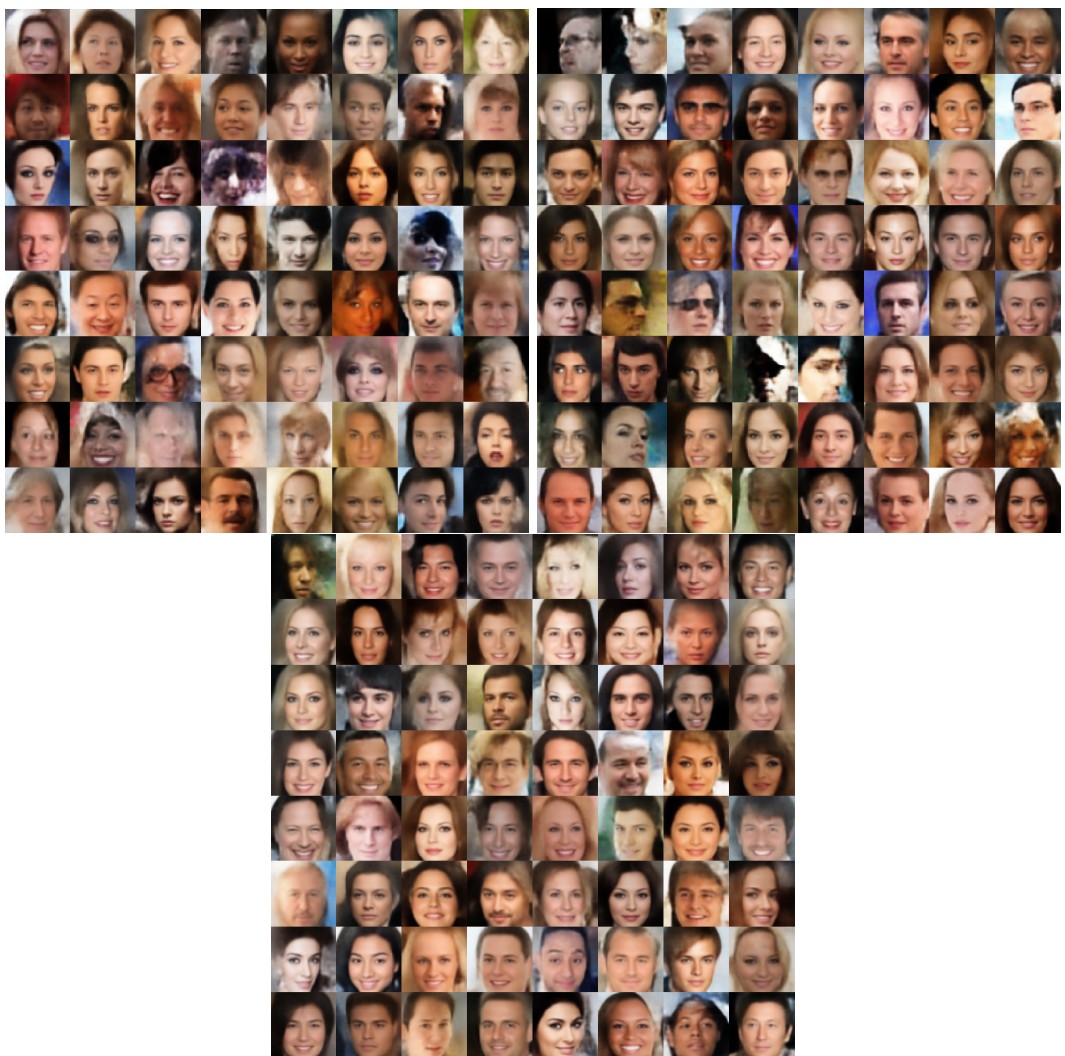

Figure 15: CelebA results from SRAE$_{\text{GMM}}$ (top left), SRAE$_{Glow}$ (top right) and our SRAE$_{\text{NTK-kPF}}$ (bottom).

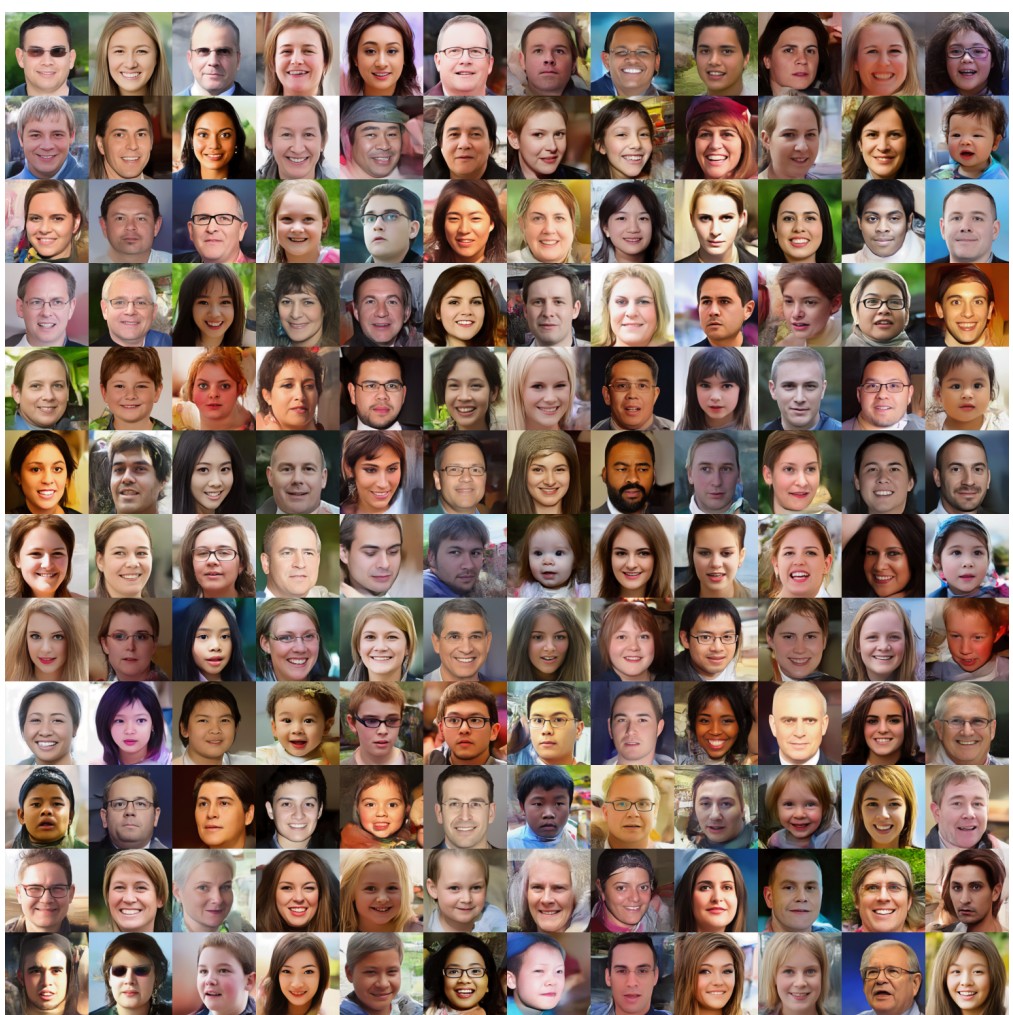

Figure 16: Additional samples from kPF+NVAE pre-trained on FFHQ

