# OpenReview forum: "Distribution Matching in Deep Generative Models with Kernel Transfer Operators"
_ICLR.cc/2022/Conference — ICLR 2022 Submitted_

### Official Review · Reviewer_wZ8o · 2021-10-26

**Correctness:** 3
**Technical Novelty And Significance:** 3
**Empirical Novelty And Significance:** 3
**Recommendation:** 6
**Confidence:** 4

**Main Review:**

Strengths:
* A conceptually simple kernel method to model densities in latent spaces of AEs.
* A simple linear operator in RKHS that maps the embedded base to the embedded data distribution, neat idea!
* Results appear to be comparable to sota methods of density modelling, while an order of magnitude faster.

Weak points:

Novelty / Presentation
Section 3 of the paper on first instance looks like the authors are presenting a novel concept here. However, the transfer operator is just the conditional embedding operator of Song et al, marginalized over the conditioning variable / applied to the mean embedding of Z, plus a pre-image estimation (see alos below). The equations are exactly the same as in e.g. Song et al 2013 (e.g. see Fig 5 in that paper).
There is a lot of equation shuffling where definitions are plugged into each other, (eq 6, eq 7, and the unnumbered eq before 6, Proposition 3.3 and its proof), that re-write definitions of the underlying operators in one form or another. I am quite skeptical about this form of presenting the method, and about the usefulness of these equations.
In addition, the connections to dynamical systems (all of which are reviewing external literature) do not appear to be necessary for the rest of the paper, especially considering Song's work, which is essentially the RKHS version of the standard rules of conditional probability.

Preimages.
As the authors mention, pre-images are problematic as RKHS embeddings are not necessarily surjective. I am therefore skeptical about the deterministic pre-image estimation methods presented here, especially for sampling. What if the distribution represented by \mu_X is multi-modal? The same point z could be mapped to two totally different x in that case, but with this deterministic approach, it won't. This should be discussed and investigated, even if performance suggests that this is not necessarily a problem.

Computational costs.
The method needs to store, and invert a kernel matrix between the data points, which costs O(n^2) and O(n^3) respectively. This becomes prohibitive from a few thousand points. I am very surprised that this is supposedly computationally cheaper that other methods. I also think the comparison done is very vague: the kernel method is dominated by a Cholesky decomposition / matrix inversion, while other methods rely on SGD/etc. So supposedly, there is a convergence question in Fig 5. Wouldn't it be better to present the other methods on a graph time-vs-accuracy? I find it also find "additional training time" confusing, and it is not very clear what is exactly compared here.
Question: The provided code contains code for Nystrom approximations. Are these actually used in the experiments? (Seems hard to invert a 10k kernel matrix otherwise). If so this should be described/mentioned.

Pre-trained AEs.
I understand that the presented method will not work well when applied directly on high-dimensional data, so the using an AE and work in its latent space makes sense. What I do not understand however, is how this can be compared (in terms of compute and performance) to the other methods in the paper. E.g. How is the method compared to a VAE when "all models share the same encoder/decoder"? Are the other models applied on the original data, or on the AE latents? Does it make sense to apply e.g. GLOW to AE latents? It would be great to make this more clear. For the NVAE case, I am even more confused: do we first train and NVAE, then use the kernel method in its latent space, and then compare it to the NVAE that we trained in the first place?

Brain data experiment.
It is a bit unclear why generating samples from this dataset would be desirable, how these samples can be "scientifically meaningful", and what the results show (e.g. colourbar is not annotated, etc). Some more motivation would help.

Finally, comparisons to GANs would be appropriate for the way the method is evaluated (only samples, density only shown for the toy examples)


Minor:
* Double period p2 "deep generative models and dynamical systems.."
* Inconsistent pre-trained vs pretrained.
* Typo in Intro "we often approaches this problem"
* "Use mean embedded operator" -> "Use mean embedding operator"
* p1 "the lower bound of likelihood" -> "a lower bound of the likelihood"
* many times citations are not properly punctuated, i.e. citep should be used
* The tensor-product operator in eq 1 is not defined.
* "Let us temporarily assume Z=X" ... this is never resolved.
* What does point (b) mean "The training scheme directly learns the mapping from the prior rather than approximation" ?

**Summary Of The Paper:**

The paper proposes a density modelling approach that is based on applying conditional embedding operators to RKHS embeddings of a base distribution.
The approach, thanks to use results from the kernel literature, is conceptually simpler and appears less computation heavy compared to VAE/flow models, while at the same time performing favorably.

**Summary Of The Review:**

Great idea, though there are some issues with presentation and experiments. Not very convincing as is, but a lot of potential to be improved.

---

> ### Author Response · Authors · 2021-11-15
> **Response to Reviewer wZ8o (Part 1/3)**
>
> Dear Reviewer wZ8o,
>
> Thank you for your valuable feedback on our paper. We provide answers to the comments below and are happy to clarify any additional points that may come up.
>
> ### (a) Novelty / Presentation Section 3 of the paper on first instance looks like the authors are presenting a novel concept here. However, the transfer operator is just the conditional embedding operator of Song et al, marginalized over the conditioning variable / applied to the mean embedding of Z, plus a pre-image estimation.
>
> In Song et al., one of the primary use of the operator was to predict the trajectory of a dynamical system via MAP inference and that paper is definitely central to our work. The high level idea we utilize is also nicely described in Klus et al. We hope the reviewer will agree that we credit these results numerous times throughout the paper and point out explicitly in Section 1 that we simply make use of the known kernel transfer operator results but in the context of a problem setting where it has not been studied. We view this as a key strength of the paper.
>
> ### (b) The equations are exactly the same as in e.g. Song et al 2013 (e.g. see Fig 5 in that paper). There is a lot of equation shuffling where definitions are plugged into each other, (eq 6, eq 7, and the unnumbered eq before 6, Proposition 3.3 and its proof), that re-write definitions of the underlying operators in one form or another. I am quite skeptical about this form of presenting the method, and about the usefulness of these equations.
>
> We intentionally kept the expressions/notations as similar as possible to other works, again to reiterate that we were not presenting a novel concept. Instead, the main message of the paper is that simple adjustments of these expressions alone are enough to get us most of the way in replacing the expensive training for generative tasks with a simple principled kernel approach. kPF happens to meet our needs very well (one will not be able to derive a better operator in the linear RKHS regime w.r.t. the RKHS norm of the difference, i.e., the mean maximum discrepancy between the model and data distribution), and we show empirically that this routine can indeed match or outperform other deep methods with a reduced computation cost, particularly in the few-sample regime.
>
> While the relationship with flow-based generative models (and other approaches) to dynamical systems has been pointed out in the literature, the hypothesis that a direct instantiation of kernel transfer operators can viably work on a subset of tasks in _any_ competitive way was not obvious to us at all. This is the core value proposition of this work and why we think that our findings are interesting for the community.
>
> We will appreciate suggestions if there are specific equation manipulations that are redundant and can be omitted -- these may indeed be repetitive for the expert but were intended to make the description easier to follow and self-contained.
>
> ### (c) In addition, the connections to dynamical systems (all of which are reviewing external literature) do not appear to be necessary for the rest of the paper, especially considering Song's work, which is essentially the RKHS version of the standard rules of conditional probability.
>
> We mostly agree with the reviewer. Studying the link to dynamical systems was the starting point of our development. To the reader unfamiliar with this literature, we assumed that this discussion will at least provide the context for why evaluating the relevance of transfer operators (e.g., Perron Frobenius/Koopman) and their kernelized variants as a simplification strategy is not radical or ad hoc. If this explanation is not satisfactory, we will happily compress this subsection.
>
> ### (d) The method needs to store, and invert a kernel matrix between the data points, which costs O(n^2) and O(n^3) respectively. This becomes prohibitive from a few thousand points. I am very surprised that this is supposedly computationally cheaper that other methods.
>
> We have two well-tested approaches to reduce the time/space complexity of kPF. The first is the fast pseudo-inverse introduced in appendix E, which leverages fast matrix multiplication on GPU and was used in the NVAE experiment, but it still requires storing the full kernel matrix which can be infeasible on commodity hardware. The other is a standard Nystrom approximation, which is included in our code as the reviewer mentioned. We verified that in most of our experiments that the models using the Nystrom-approximated operators achieve FIDs close to the full operators with just 1000 landmark samples, and we will be happy to include a more detailed analysis in the appendix. We also would like to note that simply solving a kernel matrix inverse on GPU in a vanilla kPF without any acceleration tricks on a reasonably large subset of points (10k in our case) remains much cheaper than SGD with neural networks.

---

> > ### Author Response · Authors · 2021-11-15
> > **Response to Reviewer wZ8o (Part 2/3)**
> >
> > ### (e) I also think the comparison done is very vague: the kernel method is dominated by a Cholesky decomposition / matrix inversion, while other methods rely on SGD/etc. So supposedly, there is a convergence question in Fig 5. Wouldn't it be better to present the other methods on a graph time-vs-accuracy? I find it also find "additional training time" confusing, and it is not very clear what is exactly compared here.
> >
> > We thank the reviewer for the suggestion. Indeed, for methods relying on iterative updates, a training time vs accuracy plot makes more sense. By ‘additional training time’, we refer to the training time of the latent space models _after_ completion of training the AE. Both changes are simple and we will clarify them in the revision. In the current submission, we reported the training time from the SRAE-based models in Table 1 (we will add SRAE_{VAE} to the revised draft). We train each model on the latent representations for a sufficient number of steps (10000 for GLOW and 50000 for VAE) and observe that the loss remains mostly unchanged for at least >500 steps. It is clear that even with this potentially prolonged training time, VAE and GLOW were still not able to match kPF in terms of FID. Therefore, the statement that our method is much more time-efficient compared to the deep alternatives holds without question.
> >
> > ### (f) Question: The provided code contains code for Nystrom approximations. Are these actually used in the experiments? (Seems hard to invert a 10k kernel matrix otherwise). If so this should be described/mentioned.
> >
> > All metrics reported in the paper are by inverting the full matrix directly. But we verified in most of our experiments that the models using the Nystrom-approximated operators achieve FIDs close to the full operators with just 1000 landmark samples. In the code, we provide the Nystrom approximation implementation to minimize potential hardware issues while testing our method which could be frustrating. We agree with the reviewer that the Nystrom approximated operators should be discussed more and we are happy to add text in the revision.
> >
> > ### (g) Pre-trained AEs. I understand that the presented method will not work well when applied directly on high-dimensional data, so the using an AE and work in its latent space makes sense. What I do not understand however, is how this can be compared (in terms of compute and performance) to the other methods in the paper. E.g. How is the method compared to a VAE when "all models share the same encoder/decoder"? Are the other models applied on the original data, or on the AE latents? Does it make sense to apply e.g. GLOW to AE latents? It would be great to make this more clear.
> >
> > Thank you for your comments. We apologize for the confusion and will edit the text. To clarify the statement that “all implemented models share the same encoder/decoder”, we specifically mean the SRAEs are shared among all models trained on the latent representations. As a side note, the vanilla VAE & two-stage VAE also use the same architecture (the only difference is that the latent space is not restricted to be spherical) for the primary encoding/decoding module.
> >
> > In the paper, in terms of performance, we compared kPF to VAEs and GLOWs trained directly on the input data (e.g., Vanilla VAE, CAGLOW) as well as ones trained on latent representations (e.g. SRAE_{GLOW}, SRAE_{VAE}). So, all options were considered. We hope that this answer fully clarifies the concern.
> >
> > For the training time evaluation, we only compare between models trained on latent spaces for a fair comparison. The use of GLOW and VAE here indeed make sense as post-hoc density estimators, where the joint model can be viewed as a VAE with an alternative prior, nearly identical to [1]. We can also compare the end to end cost of training an SRAE + kPF to other deep generative methods, and it is expected to converge faster as SRAE only needs to minimize the reconstruction loss while the time for training kPF is comparatively negligible. However, we found this result to be obvious and therefore was not emphasized in the paper.
> >
> > ### (h) For the NVAE case, I am even more confused: do we first train a NVAE, then use the kernel method in its latent space, and then compare it to the NVAE that we trained in the first place?
> >
> > The reviewer is correct on the NVAE experiment, that we learn a kPF over the latent space of a pre-trained NVAE model. We will make this clear in the revised draft.

---

> > > ### Author Response · Authors · 2021-11-15
> > > **Response to Reviewer wZ8o (Part 3/3)**
> > >
> > > ### (i) Brain data experiment. It is a bit unclear why generating samples from this dataset would be desirable, how these samples can be "scientifically meaningful", and what the results show (e.g. colorbar is not annotated, etc). Some more motivation would help.
> > >
> > > Generating synthetic (in the sense that they do not correspond to real subjects) but is statistic-preserving is an important emerging use case where deep generative models have a useful role to play. This is discussed in some detail in Yoon et al (doi: 10.1109/JBHI.2020.2980262. PMID: 32167919) and Dinh et al (doi: 10.1109/TPAMI.2020.3013433. PMID: 32763848). A recent high profile use of such an approach was in making EHR data of individuals tested for novel coronavirus available to researchers, see “synthetic dataset” in the NIH announcement (https://ncats.nih.gov/n3c/about/data-overview)
> > >
> > > The results here show that our method can generate realistic and disease-signal-preserving samples even with small number of samples (<500, common in biomedical datasets), which highlights the potential use of kPF in these applications. The colorbar here indicates the thresholded log p-value. We will make this clear in text.
> > >
> > > References:
> > >
> > > [1]: Dai, B and Wipf D. Diagnosing and Enhancing VAE Models. https://arxiv.org/abs/1903.05789

---

> > > > ### Comment · Reviewer_wZ8o · 2021-11-15
> > > > **Thanks for the detailed responses**
> > > >
> > > > I greatly appreciate the authors' thoughtful response to my critical points and questions.
> > > >
> > > > I agree that the authors did cite all relevant work in the paper, and I did not mean to imply otherwise. However, I feel the form of presentation suggests that the paper delivers slightly more than it does (both conceptually, and technically in the proofs). The provided clarifications here definitely improve that.
> > > >
> > > > 'Instead, the main message of the paper is that simple adjustments of these expressions alone are enough to get us most of the way in replacing the expensive training for generative tasks with a simple principled kernel approach'
> > > > It would be great if this message was made more explicit in the write up.
> > > >
> > > > 'While the relationship with flow-based generative models (and other approaches) to dynamical systems has been pointed out in the literature, the hypothesis that a direct instantiation of kernel transfer operators can viably work on a subset of tasks in any competitive way was not obvious to us at all. This is the core value proposition of this work and why we think that our findings are interesting for the community.'
> > > > That is a good point and I agree.
> > > >
> > > > 'Studying the link to dynamical systems was the starting point of our development. ....'
> > > > I think it might be better to present the work from the conditional mean embedding perspective (which I think would simplify the presentation greatly) but that might be my background (which is not on dynamical systems).
> > > >
> > > > One thing I second is reviewer hWmF's comments on slightly 'misleading' (not implying intentionally misleading!) presentation.
> > > >
> > > > Again, many thanks for the provided clarifications. It would in my opinion greatly improve the paper if they were incorporated. I will update my score.

---

> > > > > ### Author Response · Authors · 2021-11-15
> > > > > **Thank you for your feedback**
> > > > >
> > > > > We thank you for your positive feedback and support of our work! We will make sure to incorporate the changes in the revised version and upload as soon as we finish addressing the concerns of all other reviewers. We are also happy to engage in more discussions and answer any further questions or suggestions you may have.

---

### Official Review · Reviewer_mptM · 2021-11-01

**Correctness:** 4
**Technical Novelty And Significance:** 2
**Empirical Novelty And Significance:** 2
**Recommendation:** 6
**Confidence:** 3

**Main Review:**

The authors propose a new method for an important problem in machine learning. The new perspective of incorporating RKHS tools into generative models is interesting and novel. Nonetheless, I am not completely convinced that the method leads to a substantial benefit compared to existing approaches. My main comments are :
Since the method relies on interpolation using RKHS it probably can’t extrapolate beyond the support of the given distribution. Doesn’t this limit the method in terms of its ability to generate new samples? I mean here that FID should not be the only metric for evaluating a generative model; other metrics are more meaningful in that sense, for example, in terms of the diversity of the generated samples.
-The authors demonstrate that their method has low runtime compared to several baselines; however, some methods are missing from these evaluations—namely, Vanilla VAE and GMM.
-On the same point, what is the inference time required by the method to generate a new sample? I think it should be longer than of a vanilla VAE.
-Can the method be used for extremely large datasets? The kernel inverse calculation should take long and also is limited by the memory restrictions.
Minor comments:
-we often approaches-> we often approach
-by via -> remove by
-SGD mentioned before it is defined.
-the eq. above eq. 2 ends with a comma, should be period.
-period missing after eq. 2.
-kPF mentioned in caption 1, before it is defined.
-The caption in figure 1 doesn’t provide a clear explanation for the proposed scheme, this should be extended.
-”A natural extension of Klaus et al.”-> perhaps you mean “A natural extension of the work of Klaus et al.”
-In general some of the citations should be in brackets.
-”There, for” this sentence is not clear.
-”See Klaus” - why is See capitalized.
-X’ and x’ are not well defined.
-step 10 in algorithm 1- this is python notation; perhaps you want to use a mathematical notation?
-NVAE in the caption of figure 3 is only defined on the next page. Please rearrange.
-SRAE is also only defined after its first mention.


**Summary Of The Paper:**

The authors propose a new type of generative model. The new scheme is based on a kernel transfer operator that leads to a cheap method for distribution matching. The authors rely on rigorous theory on RKHS and propose a framework for transferring a prior distribution linearly (in RKHS) to the data distribution. The authors demonstrate that the proposed approach leads to improved approximations of observed distributions. Specifically, the new approach can lead to the generation of new images form a given distribution and requires less training time compared to existing baselines. The paper is mostly well written, and the method relies on solid justifications. The authors demonstrate the usefulness of the method and compare it to several existing methods.

**Summary Of The Review:**

To summarize, the authors address an important problem and propose a new generative model with several benefits over existing methods in the low sample size regime. The results demonstrate that the method leads to improvement in retrieval tasks. However, the technique has some limitations (namely, the inference may take a long time, and there might be memory limitations for large datasets). The experiments do not provide a complete picture of the method's capabilities, namely how diverse the new points are and whether they really differ from the original samples. To demonstrate that a generative model is useful, the authors could try to use the model to enrich imbalanced datasets and see if clustering or classification improves.
For these reasons, I recommend a weak rejection of the paper.

---

> ### Author Response · Authors · 2021-11-19
> **Response to reviewer mptM**
>
> Dear Reviewer mptM,
>
> We thank you for reading our work and providing a valuable review. We hope to address each of your concerns below and welcome the opportunity for further discussions.
>
> ### (a) Since the method relies on interpolation using RKHS it probably can’t extrapolate beyond the support of the given distribution. Doesn’t this limit the method in terms of its ability to generate new samples?
>
> If the concern is that we do not extrapolate outside the extremes of the empirical distribution of the samples (i.e., the support of the distribution), we agree that this is difficult. However, this issue affects most deep generative models to varying extents in our experience. For example, if we assume that there exists a mode with high likelihood in real-world data but is unobserved in our data, then there is also no easy way to capture this mode. In fact, our model potentially provides some additional extrapolation power, if the mode happens to lie within the convex hull of the samples. Whether that is desirable or not is debatable. In our experiments, we found that samples generated by our method remain well aligned to the data distribution.
>
> ### (b) I mean here that FID should not be the only metric for evaluating a generative model; other metrics are more meaningful in that sense, for example, in terms of the diversity of the generated samples.
>
> As far as we understand, FID also takes into account the diversity of samples (since the Fréchet distance measures divergence between covariances). However, we agree that there are other metrics that may help access the diversity directly, and therefore we will provide the PRD curves [1] in the appendix. Our findings are fully consistent with the FID results report which is reassuring. Briefly, models getting higher FIDs are also universally higher in terms of precision and recall in most cases, indicating improvement in both sample quality and diversity. We thank you for the suggestion.
>
> ### (c) The authors demonstrate that their method has low runtime compared to several baselines; however, some methods are missing from these evaluations—namely, Vanilla VAE and GMM
>
> We have actually included comparisons to Vanilla VAE and GMM as latent space models in Fig. 5. We think that the reviewer implies comparing the end-to-end training time of VAEs/flow-based generative model to our SRAE+kPF approach. In those cases, training an SRAE is indeed faster than both of those methods, since it only needs to optimize the reconstruction error and does not require any computationally expensive operations. The additional cost of learning kPF is minuscule, which usually takes just seconds, and therefore the joint model is still more efficient than training VAE or GLOW end-to-end.
>
> ### (d) What is the inference time required by the method to generate a new sample? I think it should be longer than of a vanilla VAE.
>
> Inference from kPF is slightly more expensive than a simple VAE (intuitively, VAE only needs to sample noise, while kPF applies additional operations on that noise sample). However, the overhead is not significant at all since both matrix multiplication and interpolation are extremely parallelizable and already have very mature implementations. For reference, to sample 10k new samples, VAE takes 10 seconds while the RBF kPF takes 10.5 seconds. The NTK kPF is slower for inference (~2min to sample 10k points), but it is primarily due to the lack of efficient coding in our current implementation (currently the NTK kernel matrix is computed by transferring the data to another GPU, invoking kernel in JAX, and copying the data back to PyTorch, which is horribly inefficient). Once better implementations are made, the inference overhead would not be very significant for NTK kPF.
>
> ### (e) Can the method be used for extremely large datasets? The kernel inverse calculation should take long and also is limited by the memory restrictions
>
> kPF can be used for reasonably large datasets with some approximation strategies (e.g. Nystrom approximation), which we are happy to state more explicitly by adding this sentence to the Limitations section. But because it is a kernel based method, it does not get around the restriction that for extremely large datasets (e.g. >1M samples) additional implementation strategies must be leveraged (such as subsampling, which we have already implemented for our experiments). We note, however, that we have demonstrated its strength on reasonably sized vision datasets as well as in the limited data setting (a domain where many deep methods fail), which we believe covers a fairly broad set of applications.
>
> Finally, we thank you for pointing out the typos and will edit them out in the revision. We hope this provides you a satisfactory answer to most of the concerns.
>
> [1] Sajjadi et al. Assessing Generative Models via Precision and Recall. https://arxiv.org/abs/1806.00035

---

> > ### Comment · Reviewer_mptM · 2021-11-23
> > **Response to authors**
> >
> > I have read the response and changes made to the paper. I appreciate the effort by the authors and am mostly satisfied with the changes.
> > The paper has been improved and I will update my score to reflect this.
> > One technical issue: the revised version exceeds 9 pages, this should be addressed.

---

> > > ### Author Response · Authors · 2021-11-23
> > > **Thank you for your response**
> > >
> > > We thank you for your response and for adjusting the score favorably. We are happy to hear that your concerns have been addressed in the revision. Thank you for pointing out the issue with the page limit and we have uploaded a new draft to fix it. We are also happy to answer any further questions you may have in the next discussion phase.

---

### Official Review · Reviewer_UugV · 2021-11-02

**Correctness:** 3
**Technical Novelty And Significance:** 2
**Empirical Novelty And Significance:** 2
**Recommendation:** 6
**Confidence:** 3

**Main Review:**

Strengths:
- The idea of using kernel transfer operators for generative modeling is novel and interesting, and. The empirical results on various image datasets are also quite good.

Weaknesses:
- Notation is a bit confusing at times, as some items are not defined. For example, the authors should define the tensor product operation in Definition 2.3. Additionally, \Delta_t in the integral right before Eq. (2) is not defined, and I’m not sure what z(.) is supposed to be when describing the underlying dynamics in Section 3. And why is the forward operator defined as f^* = (I + g^*)?

Questions:
- Why can we just embed Z into X as mentioned in the 1st paragraph of Section 3?

Miscellaneous/minor comments and typos:
- There are numerous typos/awkward sentence phrasings that should be cleaned up.
- “We often [approach] this” in the intro
- Citation formatting is confusing (should use parentheses)
- “We [demonstrate]” at the end of Section 1
- “MDS-based” in Section 4.1



**Summary Of The Paper:**

The paper proposes to leverage the Perron-Frobenius operator to simplify the mapping from Z -> X in latent variable generative models. Specifically, this “forward operator” can be approximated by a closed-form linear operator in the RKHS. They evaluate their method on both synthetic settings and high-dimensional image datasets such as MNIST, CIFAR-10, CelebA, and FFHQ.

**Summary Of The Review:**

This paper proposes to use kernel transfer operators for generative models, by leveraging a kernelized embedding of the “forward operator” (generative model). The authors experiment with various kernels and obtain favorable results on high-dimensional image generation tasks.

---

> ### Author Response · Authors · 2021-11-19
> **Response to reviewer UugV**
>
> Dear Reviewer UugV,
>
> We thank you for your valuable feedback. We agree that some clarity issues can be easily addressed when introducing our method, particularly with respect to the relation to dynamical systems. In our response below, we aim to address all concerns/questions in the review and welcome the opportunity to engage in further discussion.
>
> ### (a) $\Delta_t$ in the integral right before Eq. (2) is not defined. And I’m not sure what $z(\cdot)$ is supposed to be when describing the underlying dynamics in Section 3.
>
> $\Delta_t$ is the derivative of the state $z(t)$ at a given time $t$, and $z(t)$ is an intermediate solution of the ODE at time $t$, given the initial condition $z_{t0}$. We apologize for not stating this and will clarify.
>
> ### (b) Why is the forward operator defined as $f^* = (I + g^*)$?
>
> The definition in this form only serves to introduce the form of the (optimal) forward operator in the context of dynamical systems (similar to Eq. 3 in [1], where $\Delta_t(\cdot)$ in our text is equivalent to $f(\cdot, t)$ in that paper). Precisely, given an ODE defined by $\frac{d z(t)}{d t} = \Delta_t(z(t))$ and initial condition $z_{t_0}$, the forward operator is defined as the boundary condition at $t = t_1$.
>
> We wanted to make our formulation consistent with recent methods in deep implicit learning such as Neural ODE [1]. However, we do acknowledge that the explanation can be made more concise (e.g., instead of introducing the complete form of the dynamical system, we can simply use two boundary conditions to define $f^*$). We appreciate suggestions if this would be clearer.
>
> ### (c) Why can we just embed $\mathcal{Z}$ into $\mathcal{X}$ as mentioned in the 1st paragraph of Section 3?
> Thanks for bringing up this point.
> $\mathcal{Z}$ is assumed to be the space of a random variable that conforms to some known distribution (e.g., Gaussian) that we are transferring from. In our setup, we require $\mathcal{Z}$ and $\mathcal{X}$ to be the same space to remain consistent with results in Perron-Frobenius operators or transfer operators [2, 3]. However, for some deep generative models (e.g., VAE and most GANs), the noise is sampled in a low-dimensional space. In those cases, we can simply extend Z with auxiliary dimensions such that the dimensionality matches and they share the same coordinate system, which is merely used for the notation convenience.
>
> We hope that this reasoning will help show to the reader that most generative models which learn a direct mapping from noise to the data may fit into our framework, regardless of the specific architectures and training schemes. We will make sure that this message is more directly emphasized at the beginning of Section 3 and in the conclusion.
>
> Thank you for pointing out the few typos and formatting issues. We appreciate it! If there are any other specific places where the phrasing/wording may be improved, we will be happy to edit in the revision.
>
> References:
>
> [1] Chen et al. Neural Ordinary Differential Equations. https://arxiv.org/abs/1806.07366
>
> [2] Klus et al. Eigendecompositions of transfer operators in reproducing kernel hilbert spaces. https://arxiv.org/abs/1712.01572
>
> [3] Song et al. https://dl.acm.org/doi/10.1145/1553374.1553497

---

> > ### Comment · Reviewer_UugV · 2021-11-28
> > **reply to author response**
> >
> > Thank you for the detailed response. I'm confirming that I have read the rebuttal, updated version of the PDF, and the other reviewers' responses as well. The explanations cleared up some of my confusion, so I will keep my score as is.

---

### Official Review · Reviewer_hWmF · 2021-11-03

**Correctness:** 4
**Technical Novelty And Significance:** 4
**Empirical Novelty And Significance:** 2
**Recommendation:** 6
**Confidence:** 3

**Main Review:**

I'm deeply impressed by the novelty. It is a smart idea to start with the formulation of data generation as transforming a tractable prior to the data distribution, and then implement the formulation in RKHSs using a corresponding transfer operator. Up to my knowledge I did not notice similar ideas before. The technical details seem good to me. The paper also presents a satisfying introduction to involved techniques (nevertheless the paper may need to highlight the novel technical contribution).

However, the generative modeling approach is actually quite different from common ones, which leads to my two major concerns.
* The presentation seems misleading.
  First comes my understanding of the method. It is basically not of the same type of common deep generative models: it is non-parametric, it does not have a training stage, and its generation process is a manipulation on all training examples. In this sense, it is of the same type of kernel density estimation (KDE). Also, the method constructs the transfer operator by assuming the independence between $z$ and $x$. This makes $z$ not the usual latent variable: it does not hold any information of data, but would rather act as the random seed. (In analogy to KDE, this $z$ may correspond to the random noise around each data point).

  What I felt most misleading is the title. There is no distribution matching: the method constructs a transfer operator based on the distributions at the two ends, instead of matching a learnable distribution to a fixed target distribution. Also, the method itself is not a deep generative model: it is non-parametric (does not involve a parametric model), so "deep" does not make sense (the deep AE is independently pretrained).

  I may prefer presenting the method as it is and highlighting its unique utility/benefit as it is (instead of telling its benefits as a deep generative model, on which see the issues below; it could have other utility like learning-free/model-free data augmentation).

* Practical usefulness as a generative model.
  - Calculating the inverse of Gram matrix (Step 6 in Alg. 1) seems to cost a complexity cubic to the dataset size. Although Fig. 5 shows the training time comparison, I think asymptotically, increasing the dataset size may not lower the convergence time of parametric methods (as long as the data samples are iid), but the cost of the proposed method increases cubicly anyway.
  - Following the above point, in the development of generative models people are trying to explain that the model is not just remembering the training data. This type of method instead just generates by manipulating training data, at the other extreme. Is this acceptable? Particularly, the time complexity for deployment / data generation, would then depend on the training dataset size.
  - As mentioned, since $x$ and $z$ are taken as independent random variables, this $z$ cannot serve as a representation of data. (Posterior $p(z|x)$ is the prior $p(z)$. Also for this reason I did not understand the first limitation listed in Sec. 6. This is also what makes the approach in the same type of KDE.) This disallows the usage for embedding learning, manipulated generation, dimensionality reduction, clustering, etc.
  - As also mentioned in the paper, the method requires both marginal distributions to be supported on the whole space but not on a lower-dimensional manifold. When the latter is the case, an auxiliary model is required for dimensionality reduction.

I have some other minor questions.
* As I feel the method is of the same type as KDE, an experimental comparison with KDE is expected.
* I wonder how the choice of kernel and kernel bandwidths are set and how they influence the performance.
* What's the difference between the Perron-Frobenius operator in Definition 3.1 and the push-forward operator of a distribution by the same map $f$ (unnecessarily invertible)? Also, the domain of $f$ (and $\mathcal{P}$) seems better be $\mathcal{Z}$ (and $L^1(\mathcal{Z})$).
* I did not quite understand the method in Sec. 4.1. Particularly, what is $X$ and what is $X'$? Which one is the output of the inverse map?
* The generated data is a _linear_ transformation on training data samples. Is it flexible enough? Can it generate all possible samples following the data distribution?


**Summary Of The Paper:**

The paper presents a novel (or an unusual type of) generative modeling approach that is kernel-based and non-parametric. The basic idea is using an operator that maps from the RKHS of Z to the RKHS of X, so that data generation can be done by mapping the prior distribution $p_{prior}(z)$ to the RKHS of Z, applying the operator, and project down the result in the RKHS of X to the data space. The operator is constructed as the RKHS analogy of the desired conditional distribution p(x|z) (which makes $\int p(x|z) p_{prior}(z) dz = p_{data}(x)$), so that it transforms the RHKS embedding of $p_{prior}(z)$ to that of $p_{data}(x)$. However, to guarantee a desired $p(x|z)$, the operator is chosen by assuming $x$ and $z$ are independent. The operator can be estimated by samples of the joint distribution, which amounts to samples of $p_{data}(x)$ (i.e., training dataset) and samples from $p_{prior}(z)$ under the independent assumption, and the maps between the sample space and the RKHS can also be estimated. Experiments show the utility of the method for data generation for densely supported distributions.

**Summary Of The Review:**

I appreciate the novelty and I think the idea is worth being noticed. Nevertheless the presentation (story) seems a little misleading, which may not be a proper way to present the merit of the method.

---

> ### Author Response · Authors · 2021-11-19
> **Response to reviewer hWmF (1/2)**
>
> Dear Reviewer hWmF
>
> We thank you for your valuable feedback and finding our work interesting. Indeed, we hope our work can help spark (or rather rejuvenate) ideas in the non-parametric view of generative modeling with our promising empirical results. In our following response, we aim to address all your concerns and are happy to answer your further questions.
>
> ### (a) I may prefer presenting the method as it is and highlighting its unique utility/benefit as it is (instead of telling its benefits as a deep generative model, on which see the issues below; it could have other utility like learning-free/model-free data augmentation).
>
> The main message we try to convey in this paper is that the forward operators learned by deep generative models -- essentially performing distribution matching one way or another --  can be effectively approximated using a kernel-based operator. We are grateful that you liked the idea. This directly informed our title and the presentation of the method. We indeed do not aim to frame our method as a deep generative model -- instead, we propose that, if the distribution matching problem is considered in a RKHS (via the divergence in kernel mean embeddings or MMD), a known closed-form solution to the problem exists and efficiency gains are immediate. We are happy to further clarify this and make the benefits more salient in the revised draft. The data augmentation aspect is indeed very interesting, and we hope to explore it in the near term.
>
> ### (b) Calculating the inverse of Gram matrix (Step 6 in Alg. 1) seems to cost a complexity cubic to the dataset size. Although Fig. 5 shows the training time comparison, I think asymptotically, increasing the dataset size may not lower the convergence time of parametric methods (as long as the data samples are iid), but the cost of the proposed method increases cubicly anyway.
>
> Yes, this is true. In the revised version, we will include additional text regarding the use of Nystrom approximation to help reduce the cost of storing the operator to $O(nk)$ and computing the inverse to $O(k^3)$, where $k \ll n$. The reviewer is correct that even with this approximation, the complexity of computing kPF may still scale sup-quadratically with the problem size (since for larger sample size $n$ one may require a larger $k$ to obtain a good approximation). For large datasets, we have already adopted a subsampling step (e.g., for CelebA, we randomly selected 10k samples to train the kPF) to keep the cost manageable which appears to work well. And for small datasets, our method demonstrated stronger performance than other deep generative models.
>
> ### (c) Following the above point, in the development of generative models people are trying to explain that the model is not just remembering the training data. This type of method instead just generates by manipulating training data, at the other extreme. Is this acceptable? Particularly, the time complexity for deployment / data generation, would then depend on the training dataset size.
>
> Yes! In our case, it is indeed acceptable to interpolate the training data. We find that this does not lead to training set memorization and diversity is preserved in all our experiments. Nevertheless, we acknowledge that interpolating the training samples is only one way to calculate an approximate pre-image (which we used due to its effectiveness and simplicity). Other principled approaches are possible, e.g. a gradient-based iteration, to generate samples beyond the convex hull of training samples.
>
> ### (d) As I feel the method is of the same type as KDE, an experimental comparison with KDE is expected.
>
> We will incorporate our new results in generating samples from Gaussian KDE in the revised text, as suggested. Our experiments suggest that KDE is a good baseline although its performance is highly sensitive to the choice of bandwidth. For small bandwidths, KDE can achieve very high FID values, but in general it is due to memorization of the training samples (we will show specific examples in the appendix). Our kPF operators, especially with NTK as the input kernel, are generally more robust to the memorization issues and choice of kernel parameters.
>
> ### (e) I wonder how the choice of kernel and kernel bandwidths are set and how they influence the performance.
>
> We observed that the kernel parameters indeed influence the visual quality and diversity of the generated samples. We included a brief discussion regarding the choices of kernels and parameters in appendix F which we are happy to expand. To fully explore the influence of the kernel setup on the performance, we will provide an FID table for 25 different kernel instantiations in the revision.

---

> > ### Author Response · Authors · 2021-11-19
> > **Response to reviewer hWmF (2/2)**
> >
> > ### (f) The generated data is a linear transformation on training data samples. Is it flexible enough? Can it generate all possible samples following the data distribution
> > In principle, the support of the distribution learned by kPF is not restricted by the training samples. However, for the interpolation-based preimage we used, we indeed can only interpolate within the convex hull of training samples. While we acknowledge that this may introduce some hidden limitations (e.g., it can be hard to sample from the ‘tails’ of the true data distribution, which may have no or few corresponding training samples), we demonstrated through experiments that it still produces accurate and diverse samples which aligns well with the observed data distribution.
> >
> > ### (g) What's the difference between the Perron-Frobenius operator in Definition 3.1 and the push-forward operator of a distribution by the same map f  (unnecessarily invertible)? Also, the domain of $f$ (and $\mathcal{P}$) seems better to be $\mathcal{Z}$ (and $L^1(\mathcal{Z})$).
> > The PF operator defined in Sec 3.1 is in fact equivalent to the push-forward operator of the distribution (or rather the density measures) . We also do not require $\mathcal{P}$ to be invertible here. The reviewer is correct that the effective domain of $f$ is indeed $Z$ (and similarly for $\mathcal{P}$). However, we choose to embed $\mathcal{Z}$ to the space $\mathcal{X}$ (i.e. expanding it with auxiliary dimensions) in our derivation for notational convenience since in [1, 2] only self-maps are considered.
> >
> > ### (h) I did not quite understand the method in Sec. 4.1. Particularly, what is $X$ and what is $X′$? Which one is the output of the inverse map?
> >
> > $X’$ is just a set of neighbors of the transferred embedding $\Psi^*$ in X, which we cover later in the text. For both types of pre-image methods, we only consider a neighborhood of samples, which is denoted by $X’$, for reconstructing the pre-image, which is aligned with previous works [3, 4], We will move the first mention of $X’$ to further up in this section. Thank you for bringing this up.
> >
> > We hope that our response clarifies most of your concerns, and we welcome any further questions/suggestions you may have.
> >
> > References
> >
> > [1] Klus et al. Eigendecompositions of Transfer Operators in Reproducing Kernel Hilbert Spaces. https://arxiv.org/abs/1712.01572
> >
> > [2] Song et al. Hilbert Space Embeddings of Conditional Distributions with Applications to Dynamical Systems. https://dl.acm.org/doi/10.1145/1553374.1553497
> >
> > [3] Kwok and Tsang. The Pre-Image Problem in Kernel Methods. https://www.aaai.org/Papers/ICML/2003/ICML03-055.pdf
> >
> > [4] Honeine and Richard. Preimage problem in kernel-based machine learning. https://ieeexplore.ieee.org/document/5714388

---

> > > ### Comment · Reviewer_hWmF · 2021-11-21
> > > **Thanks for your reply.**
> > >
> > > Thanks for your elaborated response.
> > >
> > > (a) As you "indeed do not aim to frame the method as a deep generative model", the current title does not seem proper to describe your contribution. Also, at least personally and currently, I think the proposed method is more like to link two given distributions or to transfer one to the other, rather than to match some modeled or constructed distribution to the data distribution.
> > >
> > > (c,d) Perhaps a proper diversity measurement (e.g., the coverage and matching score for molecular conformation generation <https://arxiv.org/pdf/2102.10240.pdf>; though not very sure if it is suitable for images) could help justify the advantage over KDE and that the method is not working by memorizing all training samples.
> > >
> > > (g) Thanks for clarifying. But maybe using $\\mathcal{Z}$ is easier for readers to understand what's happening, and you may invoke the embedding technique in proofs.
> > >
> > > I'm roughly good with other reply items, but I still feel the way of presentation not the very right way, so I tend to keep my score. I'd prefer go frank with all the differences, and both the resulting advantages (novel, non-parametric, etc) and disadvantages (O(n^3) (or O(k^3)) complexity, cannot extract representation, do manipulated generation, etc).

---

> > > > ### Author Response · Authors · 2021-11-27
> > > > **Thank you for the suggestions. The revision reflects all requested changes**
> > > >
> > > > Dear reviewer hWmF,
> > > >
> > > > Thank you for your valuable feedback. We agree that the change to the title and incorporating your other suggestions (which are highlighted in the general response), although localized, much improved the overall clarity of our idea.
> > > >
> > > > If possible, please take a look at the revision and let us know of any other constructive suggestions, which we will happily incorporate. Thank you!
> > > >
> > > > Authors of paper229

---

### Author Response · Authors · 2021-11-23
**Draft updated**

Dear reviewers,

We have updated our draft and addressed all issues from your feedbacks. In summary, the changes involves:

- (__hWmF__, __wZ8o__) Title changed from "Distribution Matching in Deep Generative Models with Kernel Transfer Operator" to "Forward Operator Estimation in Generative Models with Kernel Transfer Operator", based on reviewer feedback. Since our original paper indeed builds upon approximating the forward operator, required changes in the text due to this title change is minimum.
- (__hWmF__, __wZ8o__) Included additional clarifications and explicit mentions of the strengths (non-parametric, efficient) and the weaknesses (maybe expensive to compute for large datasets, unable to perform controlled generation or clustering) in introduction and in limitations.
- (__hWmF__, __UugV__, __wZ8o__) Restructured the introduction of the dynamical system view of generative models. The forward operator is now defined directly using the boundary conditions.
- (__hWmF__, __UugV__) Extended the definitions of PF operator to mappings $f: \mathcal{Z} \to \mathcal{X}$, therefore avoiding confusions around embedding $\mathcal{Z}$ to $\mathcal{X}$.
- (__mptM__) Included additional clarification in the caption of Fig. 1.
- (__wZ8o__) Updated Fig. 5 for FID vs. time analysis. During the experiments, we discovered a mode collapse issue with the previous $\textrm{SRAE}_\textrm{Glow}$ baseline, and we have included the updated results in Tab. 2 accordingly.
- Included additional analysis for Nystr\"om approximated kPF for large datasets (__hWmF__, __mptM__, __wZ8o__), overfitting (in comparison to KDE) (__hWmF__) , sample diversity (__hWmF__, __mptM__), and different kernel configurations (__hWmF__).
- Fixed typos and errors in citation style.

We thank the reviewers again for the time on evaluating our work and encourage you to read the revised version to see if your concerns have been addressed. We also welcome further feedback in the next discussion phase and would be happy to answer further questions.

Authors of paper229

---

### Decision · Program_Chairs · 2022-01-20

**Decision:**

Reject

**Comment:**

Perron-Frobenius operator (P) is a well-known tool which maps the density of a dynamical system at time t (p_t) to that at t+1 (p_{t+1}): p_{t+1} = P p_t. The idea has recently been extended (kernel Perron-Frobenius operator (kPF); Klus et al. 2020) to map a probability measure p_Z to p_X via covariance operators (4) associated to a reproducing kernel; this corresponds to the transformation of the kernel mean embedding of Z to that of X as it is recalled in (5). The authors use the kPF technique in generative modelling to map the known prior (p_Z) to the data-generating distribution (p_X), and illustrate the idea numerically.

While the focus of the paper (generative modelling) is relevant, the reviewers had several severe issues with the submission:
1) the manuscript lacks clarity of presentation at multiple points,
2) the reviewers had concerns with the scalability of the approach (which unfortunately has not been analyzed),
3) the submission is a straightforward application of a well-established tool (kPF) in the literature; the paper lacks novelty.

Significantly more effort is required before publication.